# Architecture of gene regulatory networks controlling flower development in *Arabidopsis thaliana*

Dijun Chen [1], Wenhao Yan [1], Liang-Yu Fu [1] & Kerstin Kaufmann [1]

Floral homeotic transcription factors (TFs) act in a combinatorial manner to specify the organ identities in the flower. However, the architecture and the function of the gene regulatory network (GRN) controlling floral organ specification is still poorly understood. In particular, the interconnections of homeotic TFs, microRNAs (miRNAs) and other factors controlling organ initiation and growth have not been studied systematically so far. Here, using a combination of genome-wide TF binding, mRNA and miRNA expression data, we reconstruct the dynamic GRN controlling floral meristem development and organ differentiation. We identify prevalent feed-forward loops (FFLs) mediated by floral homeotic TFs and miRNAs that regulate common targets. Experimental validation of a coherent FFL shows that petal size is controlled by the SEPALLATA3-regulated miR319/TCP4 module. We further show that combinatorial DNA-binding of homeotic factors and selected other TFs is predictive of organ-specific patterns of gene expression. Our results provide a valuable resource for studying molecular regulatory processes underlying floral organ specification in plants.

[1] Institute for Biology, Plant Cell and Molecular Biology, Humboldt-Universität zu Berlin, 10115 Berlin, Germany. These authors contributed equally: Dijun Chen, Wenhao Yan.  Correspondence and requests for materials should be addressed to D.C. (email: chendijun2012@gmail.com) or to K.K. (email: kerstin.kaufmann@hu-berlin.de)

The transition from vegetative to reproductive growth in plants is controlled by the sequential and coordinated activity of transcription factors (TFs) that integrate environmental signals, such as photoperiod, temperature and nutrient status, and endogenous cues such as plant age. Over the past three decades, key TFs controlling floral transition and flower formation have been identified in *Arabidopsis thaliana*[1–3]. The MADS-domain protein FLOWERING LOCUS C (FLC) acts as a major floral repressor that is epigenetically downregulated during vernalization. The closely related FLOWERING LOCUS M (FLM) instead mediates the response to changes in ambient temperature[4,5]. Another flowering repressor protein, SHORT VEGETATIVE PHASE (SVP), is a MADS-domain protein that has roles in temperature responses by forming complexes with FLC and FLM-β, respectively[4,6,7]. Interestingly, SVP has a later role in floral meristem (FM) development together with two positive regulators of flowering time, the MADS-domain proteins SOC1 and AGL24 (ref. [8]). The FM identity genes *LEAFY* (*LFY*)[9] and *APETALA1* (*AP1*)[10] upregulate each other's expression in a positive feedback loop[11,12], resulting in stable specification of FM identity. *AP1* and *LFY* activate floral homeotic genes[13], which then act in a combinational fashion to specify the identities of four different types of floral organs (sepals, petals, stamens, and carpels). According to the "floral quartet model", floral homeotic proteins form organ-specific tetrameric protein complexes, and the higher-order interactions are mediated by the redundantly acting SEPALLATA (SEP) proteins (including SEP1, SEP2, SEP3, and SEP4)[14]. In this model, sepals are specified by a complex consisting of AP1 and SEP proteins, while AP1, AP3, PISTILLATA (PI), and SEP form a higher-order complex that determines petal identity. Stamen identity is specified by a tetramer that contains AP3, PI, AGAMOUS (AG), and SEP, while carpel identity is determined by a tetramer consisting of AG and SEP proteins[15]. All floral homeotic proteins, with the exception of AP2 whose classification as a homeotic factor is under debate[16], encode members of the MADS-domain TF family. AP2 was shown to repress *AG* expression in the outer floral whorls in Arabidopsis[17] as well as the expression of *AG*, *AP3*, and *PI* orthologs in Petunia[18]. Besides this, AP2 was found to regulate meristem maintenance and seed development[17,19]. The AP2 paralogs TARGET OF EAT1–3 (TOE1–3), SCHNARCHZAPFEN (SNZ), and SCHLAFMUETZE (SMZ) were identified as key repressors of floral transition[20,21].

Floral organ growth and differentiation is further controlled by other types of TFs, including BELLRINGER (BLR)[22], JAGGED (JAG)[23], ETTIN (ETT)[24,25], and REPRESSOR OF GIBBERELLIC ACID (RGA)[26]. *BLR* encodes a homeodomain TF and plays a central role in shoot apical meristem development by affecting meristem activity, phyllotaxis, and floral organ patterning[22]. JAG encodes a zinc-finger TF that regulates organ boundary formation and growth[27]. ETT is an auxin-responsive TF that regulates several aspects of flower morphogenesis[24,25], and RGA controls inflorescence development via participating in the gibberellin signaling pathway[26,28].

Chromatin immunoprecipitation (ChIP) followed by high-throughput DNA sequencing (ChIP-seq) facilitates the identification of genomic regions bound by developmental master TFs during flower development[29]. Biologists started to understand patterns of genomic TF binding[30,31], and the complex gene regulatory networks (GRNs) that govern the formation of flowers on a global scale[32]. However, a systematic analysis of the target gene networks of these master regulators is still lacking, and thus our knowledge towards the associated network components that control floral organ development is incomplete. In this regard, the combination of genomic binding and expression data provide an opportunity to shed light on the regulatory "code" underlying floral organ specification.

Many factors such as microRNAs (miRNAs) have roles in floral transition and flower development. For example, the age-dependent transition from vegetative to reproductive growth was shown to be regulated by *MIR156*, which gradually decreases with age thus results in elevated expression of its targets, *SQUAMOSA PROMOTER BINDING-LIKE* (*SPL*) TFs which are in turn upstream activators of FM identity genes, *LFY* and *AP1*, as well as *MIR172* (refs. [33–35]). *MIR172* targets *AP2* and its closely related paralogs, and thereby controls floral transition and whorl-specific activation of floral homeotic genes[16,36]. Other reports showed that *MIR319* affects petal size by regulating *TCP4* and *TCP10* genes[37]. *MIR159*, *MIR160*, *MIR164*, *MIR165*, *MIR166*, and *MIR167* were also reported to have roles in floral organ morphogenesis in *Arabidopsis*[38]. Despite many reports on miRNAs regulating floral transition and flower development, a comprehensive understanding of the dynamic control of these processes by miRNAs is still missing. In particular, we have limited knowledge about how miRNAs themselves are regulated by upstream regulators.

Here, we performed an integrative analysis of genome-wide DNA-binding data of master regulatory TFs, including factors controlling floral transition (FLC, FLM, SVP, and SOC1), FM/organ identity TFs (LFY, AP1, AP2, AP3, PI, SEP3, AG), and regulators of floral organ morphogenesis (BLR, JAG, ETT, RGA). Coupled with newly generated stage-specific mRNA and miRNA expression data, the GRN controlling reproductive growth in *Arabidopsis* was constructed and investigated with a focus on the action of miRNAs in the network. We integrated all TF-binding information with miRNA regulation into a systems-level regulatory network. We performed network analysis to systematically search for enriched network motifs in this "meta-network" and identified prevalent miRNA-mediated feed-forward loops (FFLs). We further validated an important role for an FFL consisting of *MIR319a* and its TF targets in mediating petal growth. Lastly, using a machine-learning approach, we demonstrated that combinatorial binding by homeotic regulators and other TFs to common target genes is important for stage- and organ-specific activation of gene activities.

## Results

**Landscape of TF binding during flower development**. To gain a comprehensive view of TF DNA-binding during flower development, we integrated ChIP-seq data for 15 key floral regulators, including 85 ChIP-seq datasets from 15 studies (Supplementary Table 1 and Supplementary Data 1), which can be assigned to five representative developmental stages, or to mixed tissues (Fig. 1a, b). An analysis pipeline that consists of quality control, peak calling, and target identification to re-analyze all raw data in a standardized and uniform manner was developed to obtain comparable TF-binding maps (see Methods; Supplementary Fig. 1). In this regard, the IDR (Irreproducible Discovery Rate) framework[39] was adopted as a unified approach to derive highly stable thresholds based on reproducibility. Principal component analysis (PCA) of ChIP-seq signal profiles revealed that replicate datasets, or datasets for TFs with similar functions, tend to group together (Supplementary Fig. 2). It is notable that our re-analysis recovers the majority of target genes from the correspondingly original studies (Supplementary Fig. 3a) and reveals general outperformance in terms of identifying the true TF-bound genes by integration of differential expression data upon perturbation of the corresponding TF (Supplementary Fig. 3b).

Overall, the number of TF-binding sites (TFBSs; called "peaks") varies greatly for different regulators or for the same regulator across different stages (Fig. 1c; Supplementary Data 2). BLR, LFY, and SEP3 show the highest number of binding sites, consistent with the fact that these TFs possess multiple and

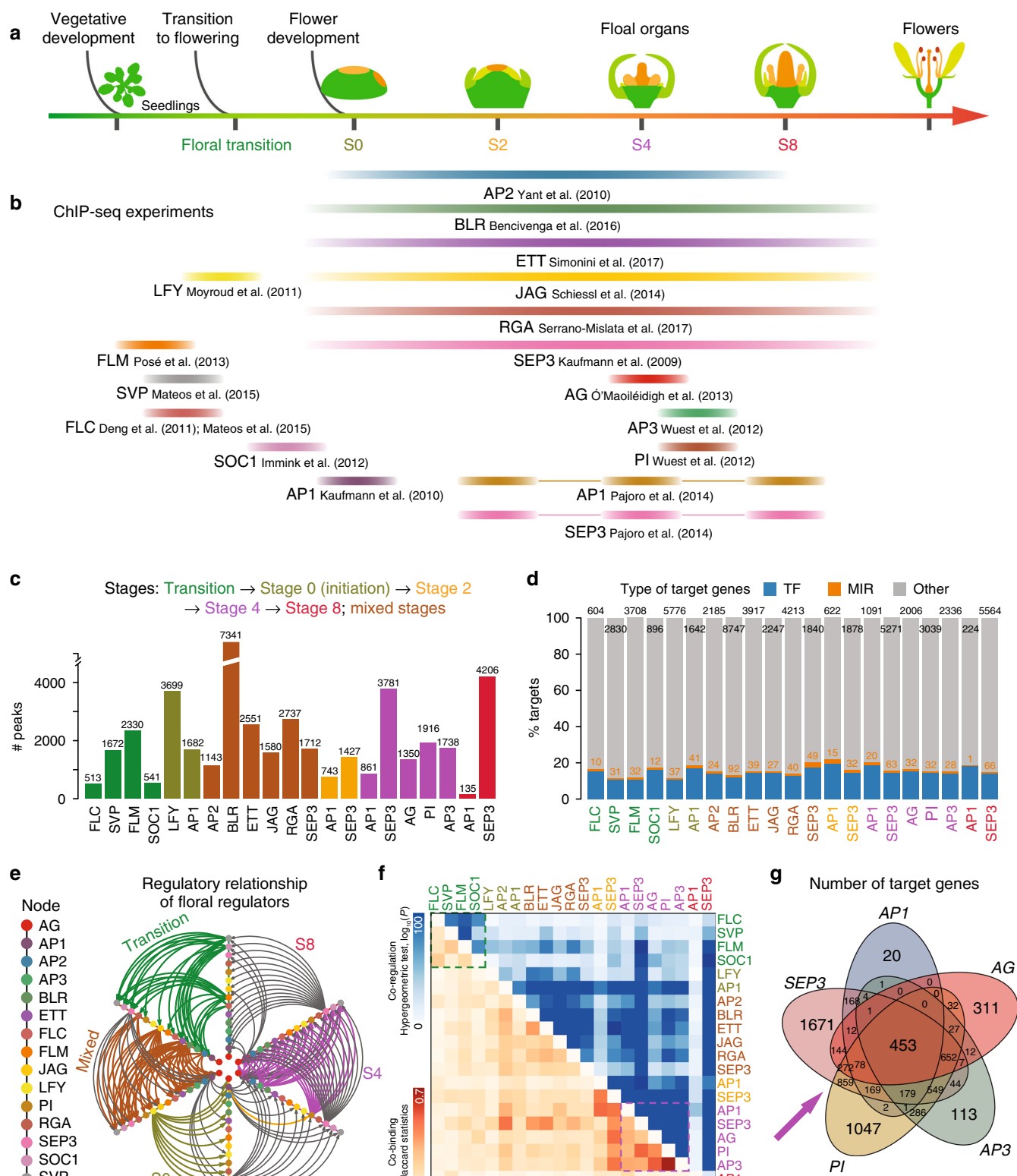

diverse roles in reproductive development[40–42]. SEP3 gains significant binding sites across flower developmental stages, in agreement with its roles during floral organ development and with its expression levels (Fig. 1c). On the other hand, AP1 loses binding sites during the developmental time-course, consistent with its important role as FM identity factor at early stages of flower development, and a more restricted expression at later time points (reducing the ChIP efficiency).

We then classified potential target genes of the investigated regulators (Fig. 1d; Supplementary Data 3) into protein-coding and miRNA gene loci. A target gene was counted only when a TF-binding event was present within 3 kb of the start of the gene. We found that most TF-binding peaks are located within 64 bp upstream of protein-coding gene TSSs (Supplementary Fig. 4a) or 313 bp upstream of miRNA precursors (Supplementary Fig. 4b). Overall, 10–20% of target genes encode TFs (Fig. 1d), and 51.4%

**Fig. 1** A DNA-binding atlas of key-regulatory transcription factors in flower development. **a** Schema depicting the key developmental stages for flower formation, including floral transition, floral initiation (stage 0, S0), meristem specification (S2), organ specification (S4), organ differentiation (S8), and flower maturation. The width of each color bar roughly defines the time range in which the plant materials were harvested for a TF ChIP-seq experiment. Note that colors representing stages are consistently used across all the figures if applicable. **b** Currently available genome-wide TF-binding maps for floral regulators. The datasets are roughly assigned to the five floral stages as defined in **a**; the source of ChIP-seq data is labeled below the bar and also indicated in Supplementary Table 1. **c** Bar chart showing the number of binding sites for the investigated TFs. TFs are colored according to the development stages. **d** Bar chart showing the percentage of target genes as defined in three classes: transcription factors (TF; blue), miRNA-related genes (MIRs; orange) and other protein-coding genes (gray). The number of target genes is shown above bars. **e** Cross-regulatory interactions between 15 TFs in regulatory networks across five stages. TFs are colored and arranged in the same order along each axis. Stage-specific edges are colored as indicated, whereas regulatory interactions present in more than one stage networks are shown in gray. **f** Co-binding relationships and co-regulated targets by TFs. Upper triangle, highly significant co-regulation relationships are assigned a dark blue color, representing $-\log_{10}(P\text{-value})$. The $P$-value was calculated from a hypergeometric test to check the significance of target overlapping between any two TFs using all the annotated genes as the background. Lower triangle, highly significant co-binding relationships are assigned a dark orange color. The significance of co-binding by any two TFs were tested by Jaccard statistics, which measures the ratio of the number of intersecting base pairs occupied by both TFs to the number of base pairs in their union. Dashed boxes indicate highly interplayed regulators in vegetative development (green) and flower development (purple). **g** Venn diagram of the target genes shared by AG, AP1, SEP3, AP3, and PI at stage 4

(167 out of 325) of the annotated miRNA genes are potentially regulated by at least one floral regulator (Supplementary Data 3). Notably, we found that the regulatory relationship among the investigated floral regulators is highly complex, including both intensive cross-regulatory interactions and auto-regulatory loops (Fig. 1e). Clustering analysis of TF-bound genomic regions and target genes revealed that the regulators are generally grouped with protein complex partners, and by the investigated stages (Fig. 1f). For example, MADS proteins controlling floral transition (i.e. FLC, FLM, and SVP) have similar binding patterns, suggesting that they may interact with each other to exert their functions in the flowering pathway[7,43,44]. Similarly, there is a high overlap of binding sites of the MADS proteins in the AG-SEP3-AP3-PI (stamen) heterotetramers (Fig. 1f, g).

**Occupancy of multiple TFs predicts gene expression dynamics**. Flower development is a dynamic process in which floral organs are produced from pools of stem cells residing in meristems[45]. In order to obtain a high-resolution map of the changes in gene expression, an inducible system to synchronize early stages of flower development[46] was used to collect stage-specific floral tissues (0, 2, 4, and 8 days after floral induction) for transcriptome profiling by mRNA-seq (Supplementary Fig. 5; see Methods for details). These stages represent the status of inflorescence meristem, FM specification, floral organ specification, and floral organ differentiation, respectively. In total, 7416 genes (representing 27.3% of the total protein-coding genes) were differentially expressed (DE) with at least two-fold change across the four developmental stages (false discovery rate [FDR] <0.05; Supplementary Data 4). The DE genes significantly overlapped with potential floral regulator target genes as determined by ChIP-seq ($P < 1.1 \times 10^{-91}$, hypergeometric test; Fig. 2a) and showed significantly positive or negative correlations between expression changes and TF-binding intensities (Supplementary Figs. 6 and 7), suggesting that these regulators control transcriptome dynamics during flower development. Generally, the DE genes, including ten of investigated regulators, were gradually either down- or upregulated across development stages (Fig. 2b). Genes targeted by more regulators tend to show significantly higher expression dynamics (Fig. 2c). To confirm the functional consequences of dynamic TF binding, we compared the expression and biological functions of genes grouped by the number of bound TFs. Flower-related GO terms were thus associated with up- or downregulated genes grouped by the number of targeting TFs (Fig. 2d). Upregulated genes during flower development are enriched for more flower-related functions with an increased number of bound TFs. For example, 66.0% (95 out of 144) of DE genes bound by more than ten TFs were upregulated, including well-known floral regulators (AP1, AP3, SEP3, and SHATTERPROOF2 (SHP2)), gibberellin pathway genes (e.g. GA INSENSITIVE DWARF1B (GID1B) and RGA-LIKE 2 (RGL2)), and light signaling genes (PHYTOCHROME A (PHYA) and PHYTOCHROME-INTERACTING FACTOR 5 (PIF5)).

To further characterize the potential common TF target genes, we grouped them by the number of bound TFs (Fig. 2e). Interestingly, a genomic cluster consisting of three closely spaced C-REPEAT/DRE BINDING FACTOR (CBF) genes was identified among the most frequently bound genes. CBF1 to 3 act as key regulators of cold acclimation in Arabidopsis[47], but are only weakly expressed in flowers. The data suggest that these genes need to be stably repressed during the reproductive phase, at least under standard conditions, via the action of multiple upstream regulatory TFs. Additionally, loci of known miRNA genes, including MIR159a, MIR168a, MIR172a/d, and MIR399a, were found to be occupied by at least 8 out of the 15 regulators (Fig. 2e).

In summary, genes bound by multiple TFs show more dynamic expression during flower development. Different types of regulatory genes and a defined set of miRNAs are among the hub target genes in the flower developmental network.

**miRNA-mediated GRNs and enriched network motifs**. Among the targets of floral master regulators, miRNA genes were frequently identified (Fig. 1d) and often bound by multiple TFs (Fig. 2e). Given that a major category of plant miRNA targets consists of TFs[48] and those targets may also be directly regulated by floral master regulators, it is appealing to elucidate developmental GRNs that are wired by floral key-regulatory TFs, miRNA genes, and their target genes. In such a "meta-network", the effect of different regulatory mechanisms can be determined, and miRNAs acting as middle players may confer additional layers of network robustness[49]. Indeed, there are in total 422 predicted TF-miRNA regulatory relationships and 273 miRNA-TF post-transcriptional interactions among the 15 master regulators, 144 expressed miRNA genes, and 674 expression-changed TF genes (Fig. 3a). In silico analysis of the network structure revealed that removal of hub genes leads to more rapid loss of interactions in the network without miRNA regulation, compared to miRNA-mediated networks (Fig. 3b). This observation suggests that miRNA-mediated regulation enhances the topological robustness[50] of the floral GRN.

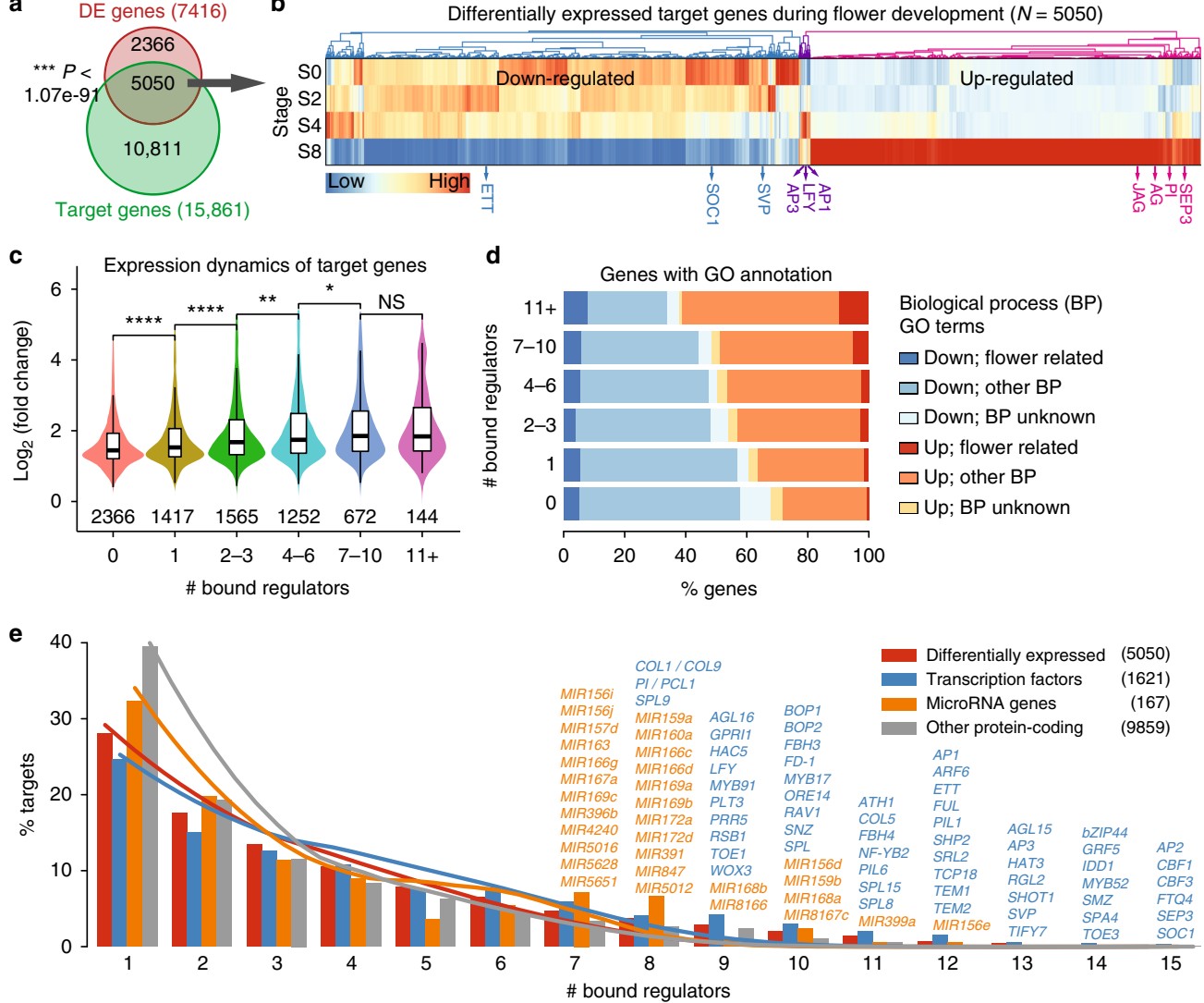

**Fig. 2** Expression dynamics and functions of target genes regulated by floral regulators. **a** Venn diagram showing significant overlap (P-value calculated by a hypergeometric test) of target genes by floral transcription factors (TFs; green) and differentially expressed (DE) genes across flower development (red). **b** Clustering analysis of DE target genes. Expression values are center normalized per gene based on its $\log_{10}$-transformed FPKM values in the four representative stages. The differentially expressed regulator genes are labeled. S0: stage 0. **c** Boxplot showing the distribution of expression changes (the maximum value of absolute fold change over stages) with increasing number of bound TFs. The number of genes in each category is labeled below the box. Boxes represent quartiles, center lines denote 50th percentile, and whiskers extend to most extreme values within 1.5× interquartile range (IQR). Significance codes, ****: P-value <0.0001, ***: P-value <0.001, **: P-value <0.01, *: P-value <0.05, and NS: not significant, by two-sided Mann–Whitney tests. **d** Expression and functional characteristics of DE genes grouped by the bound TFs. Up- and downregulated DE genes are indicated in **b**. BP: biological process. **e** Distribution of bound regulators per potential target gene. Curves illustrate the ensemble of 100 randomized networks of the same size and degree distribution. Examples of hub target genes (bound by seven regulators or more) are labeled

Apart from global network structure, we next sought to investigate the floral GRN (Fig. 3a) from the perspective of network motifs, which represent simple regulatory circuits that carry out specific biological functions[51]. To simplify, three different network motifs, namely, one-node (autoregulation), two-node (feedback loop), and three-node motifs, were considered (Fig. 3c). These regulatory scenarios reflect the basic topology of GRNs. Autoregulation is the simplest but important motif commonly found in networks exhibiting multi-stability[52] and was found for 11 out of 15 regulators (Fig. 3d). Feedback loops represent regulation between different master regulators (n = 34; see also Fig. 1e) or between a master regulator and its miRNA target genes, e.g., the feedback regulation between MIR172 and AP2 (refs. [20,53]) (Fig. 3e). Furthermore, we

systematically searched all possible three-node motifs and then determined which of these were significantly over- or under-represented relative to a random control[54] (Fig. 3c). The most abundant and enriched three-node motif is the single-input module (SIM). We enumerated all possible instances of a floral master regulator regulating two TFs and found that the master regulators are more likely to regulate a pair of physically interacting factors (Fig. 3f), indicating that a master regulator tends to control an entire functional unit (i.e., a TF complex) rather than just a single component[55].

FFL motifs are particularly interesting because they describe the targeting of a master regulator to both miRNAs and miRNA targets. Such trifurcate regulatory circuits theoretically provide more flexibility and pliability for a master regulator to achieve

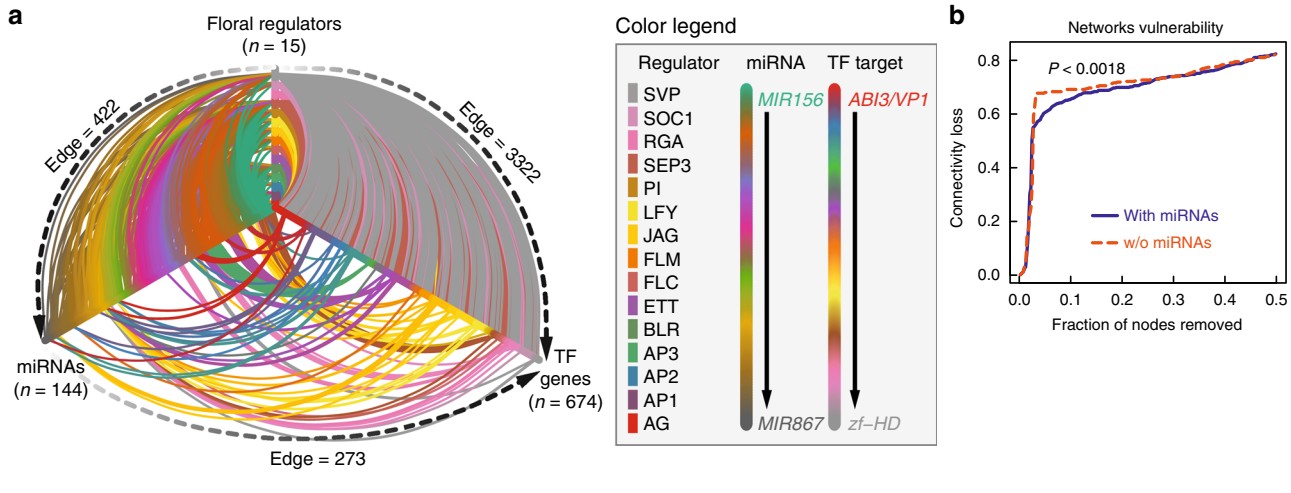

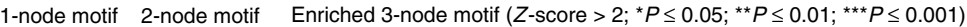

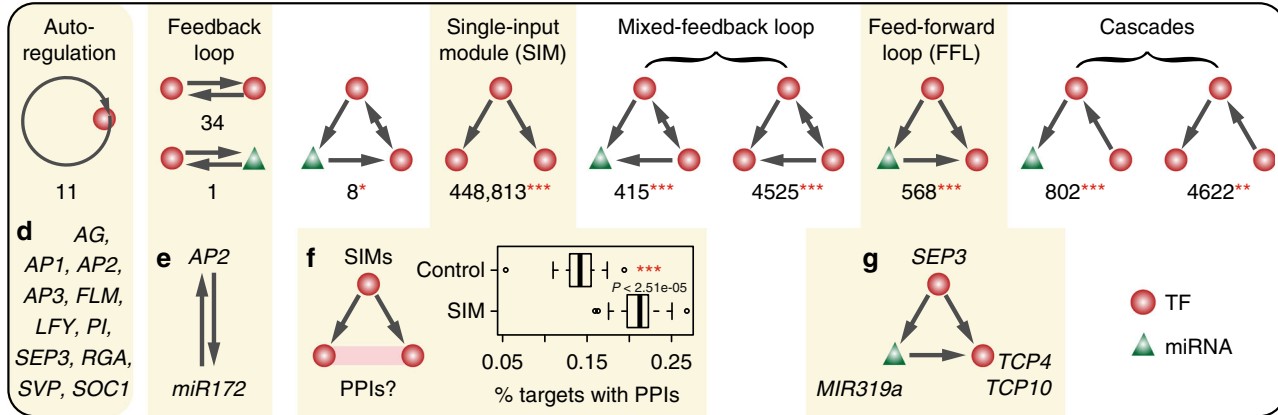

**Fig. 3** Characterizing flowering gene regulatory networks and motifs. **a** Hive plot showing gene regulatory networks involved in floral regulators, miRNA genes, and transcription factors (TFs). Connections indicate potential regulation relationships. Regulator-miRNAs and regulator-TFs relationships are supported by ChIP-seq data, while miRNA-TFs relationships are based on prediction. For valorization purpose, nodes are colored according to TF or miRNA gene families or regulators, and edges are colored according to the corresponding source or target. Note that only differentially expressed TF target genes are shown. **b** Robustness of miRNA-mediated gene regulator networks. Plot shows the connectivity loss due to iteratively removing nodes with the maximum load (betweenness) in the networks with or without (w/o) miRNA-mediated connections. P-value was calculated based on a two-sided Kolmogorov–Smirnov test. **c** Systematic search of one-node, two-node, and three-node motifs from the network represented in **a**. The number of motif occurrences is shown below the schema. For three-node motifs, only statistically enriched (significance code denoted in parenthesis) motifs are shown. Examples of corresponding motifs are listed at the bottom panels **d**–**g**. **d** Autoregulation as the simplest motif consisting of the master regulators themselves. **e** The feedback regulation between *MIR172* and *AP2*. **f** Analysis of the single-input module motifs (SIMs). A floral regulator may target two TFs that are connected via protein–protein interactions (PPIs; left schema). The right boxplot shows the percentage of target genes with PPIs. As control, the same number of target genes were sampled from the whole target gene list and the percentage of targets with PPIs were calculated. P-value was calculated based on a two-tailed Student's t-test. Boxes represent quartiles, center lines denote 50th percentile, and whiskers extend to most extreme values within 1.5× interquartile range (IQR). **g** A feed-forward loop (FFL) motif mediated by *MIR319a*

fine tuning of downstream gene expression[49,56]. Examples of this motif have been identified in recent work on leaf age-dependent cell death[57] and the adventitious rooting program[58] in *Arabidopsis*. Interestingly, we predicted 568 FFL motifs involving 57 distinct miRNA genes and 88 TF genes (Fig. 3c; Supplementary Data 5). We further focused on potential AP1- and SEP3-regulated FFLs, for which the corresponding time-series gene expression data and TF-binding data were both available in order to investigate the dynamics and functional consequence of FFL regulation. In total, 135 AP1- and SEP3-regulated FFLs were predicted (Supplementary Fig. 8). We experimentally validated dynamic expression changes for several genes (Supplementary Fig. 9a) and observed a significantly positive correlation between the experimental data and the sequencing data (Pearson's

correlation coefficient $r = 0.51$ and $P < 0.001$; Supplementary Fig. 9b). The expression change of a target TF gene could be due to either the expression change of its miRNA regulator(s) or the binding change of its upstream master regulator(s). We found that 73.3% (99/135) of the AP1- and SEP3-regulated FFLs were supported by significant changes of either miRNA gene expression or TF-binding level (Supplementary Data 6), thus were considered as developmentally dynamic loops. A major question would then be whether the binding of a master regulator to both miRNA and its target TF loci has a biological function. To explore this point, we sought out to characterize a developmentally dynamic regulatory circuit in which SEP3 was found to target both *MIR319a* and its putative targets, *TCP4* and *TCP10* (Fig. 3g).

**SEP3 as a direct regulator of *MIR319a*.** It has been shown that ectopically expressed *MIR319a* (*jaw-D* mutant) generates larger petals, while a *MIR319a* loss-of-function mutation (*miR319a*[129]) results in narrower and smaller petals compared to wild type. These phenotypes are mainly due to alteration of the *MIR319a*-TCP4 regulatory module[37]. However, how *MIR319a* expression is regulated is largely unknown. In our analysis, SEP3 ChIP-seq peaks were found in the upstream regions of *MIR319a*, *TCP4*, and *TCP10* genes (Fig. 4a), indicating a possible regulation role of SEP3 in regulation of *MIR319a* and TCP genes. A larger petal size that had been observed in the *jaw-D* mutant was also observed in the *sep3-2* single mutant (Supplementary Fig. 10a–c). A line with enhanced expression of *SEP3-GFP* driven by a 4.1 kb endogenous promoter (SEP3[HE4.1k]) containing a distal enhancer element[59] showed a similar floral phenotype to the one observed in plants with *MIR319a* loss-of-function (Fig. 4b, c). More strikingly, the SEP3[HE4.1k] line largely rescued the petal size defects of *jaw-D* mutant in which *MIR319a* is overexpressed (Fig. 4b, c). These observations suggest that *SEP3* and *MIR319a* have opposite roles in petal growth. Accordingly, the expression of *MIR319a* was downregulated in SEP3[HE4.1k] lines, and the expression of both *MIR319a* and its targets (i.e., *TCP4* and *TCP10*) recovered to wild-type level in plant lines that combine the *jaw-D* mutation and SEP3[HE4.1k] (Fig. 4d). Since floral organ size is largely determined by the timing of the transition from cell division to cell expansion, which occurs at around stage 10 of flower development[60], we analyzed expression of *TCP4* and *MIR319a* in flowers of both SEP3[HE4.1k] and wild-type flowers, with the material harvested around floral stage 10. Interestingly, we observed clearly elevated *TCP4* expression in SEP3[HE4.1k] lines but comparable expression of *MIR319a* in SEP3[HE4.1k] lines and wild-type plants, and the difference in *MIR319a* expression persisted when using whole inflorescences (Fig. 4e). Together, the results indicate that the binding of SEP3 to the *MIR319a* locus acts to dampen elevated *MIR319a* levels (as observed in the jaw-D/SEP3[HE4.1k] double mutant), while SEP3 positively regulates *TCP4* expression to restrict petal size. To further confirm a direct regulatory effect of SEP3 binding to *MIR319a* and *TCP4*, transient assays using a dual-luciferase reporter system were performed. The results showed that deletion of SEP3 ChIP-seq peak in the *MIR319a* promoter, and of the far-upstream SEP3 ChIP-seq peak in the *TCP4* promoter could significantly reduce the effect of SEP3 on luciferase expression (Supplementary Fig. 11). In all, these results show that the binding of SEP3 to both *MIR319a* and *TCP4*, together with the targeting of *MIR319* to TCP genes, form a coherent FFL to regulate petal growth in *Arabidopsis* (Fig. 4f).

**Organ-specific GRNs.** A recent study[61] showed that DNA-binding specificity of homeotic MADS-domain proteins contributes to organ-specific gene regulation. We asked how floral organ GRNs become specialized through combinatorial control by floral homeotic proteins and other TFs that function in floral organ morphogenesis and growth. Here, we excluded vegetative MADS factors (FLC, FLM, SOC1, and SVP) from the analysis. For AP1 and SEP3, only binding data at stage 4 were considered. Therefore, binding maps of 11 TFs were included in subsequent analysis. Based on domain-specific mRNA translatome data[62], we identified 1013 organ-specific genes that were highly expressed either in AP1-specific (sepal), AG-specific (carpel), AP1/AP3-common (petal), or AP3/AG-common (stamen) domains (see Methods). Of note, 66.9% (678 out of 1013; $P < 2.2 \times 10^{-16}$) of these domain-specific genes were bound by at least 2 of the 11 selected TFs (Supplementary Data 7). Clustering analysis based on gene expression data revealed groups of genes that are specific to domains of homeotic gene activities (Fig. 5a). For

example, *SPOROCYTELESS* (*SPL*)[63], a master regulator of gametogenesis, was found to be specific to AG and AP3 domains, while the petal number regulator *PETAL LOSS* (*PTL*) is specific to AP1 and AP3 domains. Coordinated presence of TFBSs from 11 TFs that act in flower morphogenesis can largely predict the floral domain-specific expression patterns (Fig. 5b). To identify potential regulators affecting domain specificity, we built a regression model of expression changes between two different domains using normalized peak signals as features. The resulting model predicted that homeotic MADS proteins and other TFs act in concert to determine whorl-specific expression patterns. In detail, ETT and BLR contribute to expression differences between perianth (AP1) and reproductive (AG) organ whorls, and AP2 appears to be important for determining expression differences between the outer whorls and carpels (Fig. 5c). Other factors, such as LEAFY (LFY) and JAGGED (JAG), are predicted to contribute less to organ-specific gene regulation.

To further investigate organ-specific GRNs, we used ChIP-seq data to predict 167 target genes that act downstream of the homeotic TF complexes to specify sepals, petals, stamens, or carpels based on the floral quartet model (Fig. 5d). We visualized these genes in a network in which nodes represent domain-specific gene expression patterns (Fig. 5e). As expected, genes that were implicated in sepal and petal networks tended to have higher gene expression levels in AP1 and/or AP3 domains, whereas genes in stamen and carpel sub-networks showed higher expression in AP3 or AG domains. Using this approach, we were able to recover sets of organ-specific vs. common target genes, as well as to predict genes that are modulated in an organ-specific manner to generate the distinct morphologies of different types of floral organs. Those genes showing floral domain-specific expression patterns include *CUP-SHAPED COTELYDON1* (*CUC1*) and *CUC2* (ref. [64]), *BLADE-ON-PETIOLE 1* (*BOP1*) and *BOP2* (ref. [65]), as well as the F-box gene *STERILE APETALA* (*SAP*) and its target *PEAPOD2* (*PPD2*)[66]. *CUC1* and *CUC2* are redundantly acting factors required for establishing boundaries of floral primordia[67,68]. Similarly, *BOP1* and *BOP2* are expressed in the boundaries of floral primordia, and they regulate FM identity[69]. Both *CUC1/2* and *BOP1/2* genes are preferentially expressed in the AP3 domain, whereas *SAP* and *PEAPOD2* (*PPD2*) showed elevated expression in the AG domain (Fig. 5e). SAP and PPD2 are known to regulate flower organ size and fertility through an E3 ubiquitin ligase-mediated protein degradation pathway[66]. Therefore, this analysis serves as a starting point for a more systematic genetic analyses of regulatory links that govern floral organ morphogenesis.

## Discussion

Flower morphogenesis is a highly robust and standardized developmental patterning process that is initiated by the combination of environmental and endogenous signaling pathways. While it has been well studied that master regulatory TFs that act at different stages of the flowering process form complex regulatory loops, their regulatory interplay is much less well understood. In particular, organ-specific GRNs of floral homeotic proteins remained enigmatic, since only a small fraction of potential direct target genes of these factors was found to be DE in flower development. Here, we show that systematic combination of DNA-binding data of 15 TFs with regulatory roles in flower development is able to predict genes that are dynamically regulated during flower morphogenesis. The resulting GRN is enriched for transcription regulators and for miRNA loci. The network structure shows a particular overrepresentation of the well-studied FFL motif that is mediated by the combined activity of TFs and miRNAs.

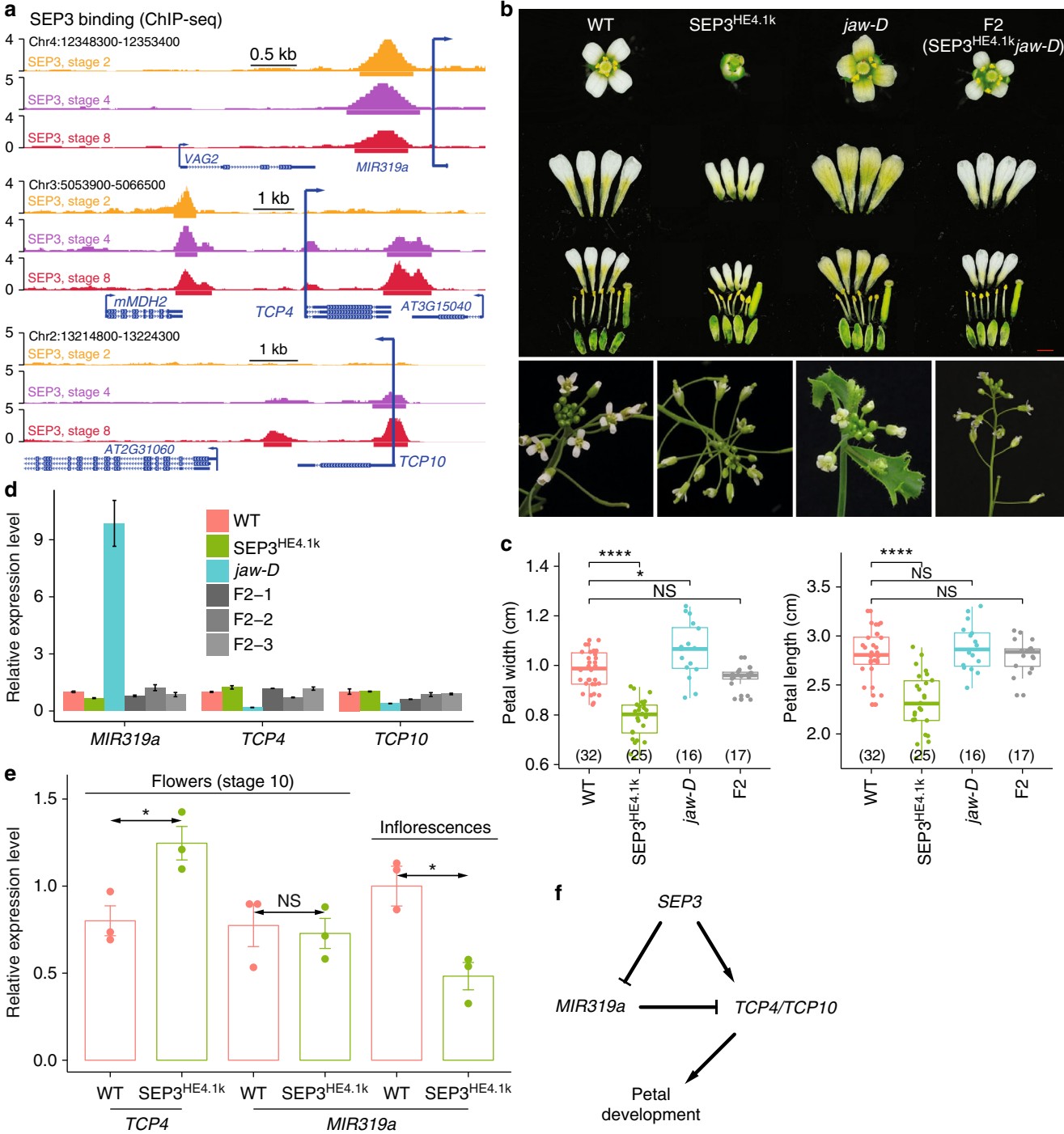

**Fig. 4** SEP3 as a direct regulator of *MIR319a*. **a** Dynamics of SEP3 binding at the loci of *MIR319a* (top), *TCP4* (middle), and *TCP10* (bottom). Arrows indicate transcriptional directions. **b** Pictures of single flowers and separated petals as well as other flower organs (top panel; scale bar = 1 mm) and inflorescences (bottom) of wild-type (WT) plants, enhanced expression of *SEP3* (SEP3^HE4.1k), *jaw-D* mutant lines, and combination (F2) of *jaw-D* mutant and SEP3^HE4.1k. **c** Boxplots showing the petal width (left) and petal length (right) of different genotypes in **b**. The number of biologically independent plants is shown in parentheses. Boxes represent quartiles, center lines denote 50th percentile, and whiskers extend to most extreme values within 1.5× interquartile range (IQR). Significance codes, ****: *P*-value <0.0001, *: *P*-value <0.05, and NS: not significant, by two-sided Mann–Whitney tests. Bar = mean ± s.e. **d** Relative expression of *MIR319a*, TCP4, and TCP10 in the inflorescences of WT, *jaw-D* mutant, SEP3^HE4.1k lines, and three independent F2 lines produced by crossing *jaw-D* and SEP3^HE4.1k lines. **e** Relative expression of *TCP4* and *MIR319a* in either whole inflorescences or stage 10-enriched flower tissues of WT and *SEP3* enhanced expression lines. For all the expression analysis, data represent mean of three independent biological replicates. The *Tip41* gene (AT4G34270) was used as reference. Significance codes as in **c**. **f** *SEP3-MIR319a-TCP4/TCP10* coherent feed-forward loop in regulation of petal development

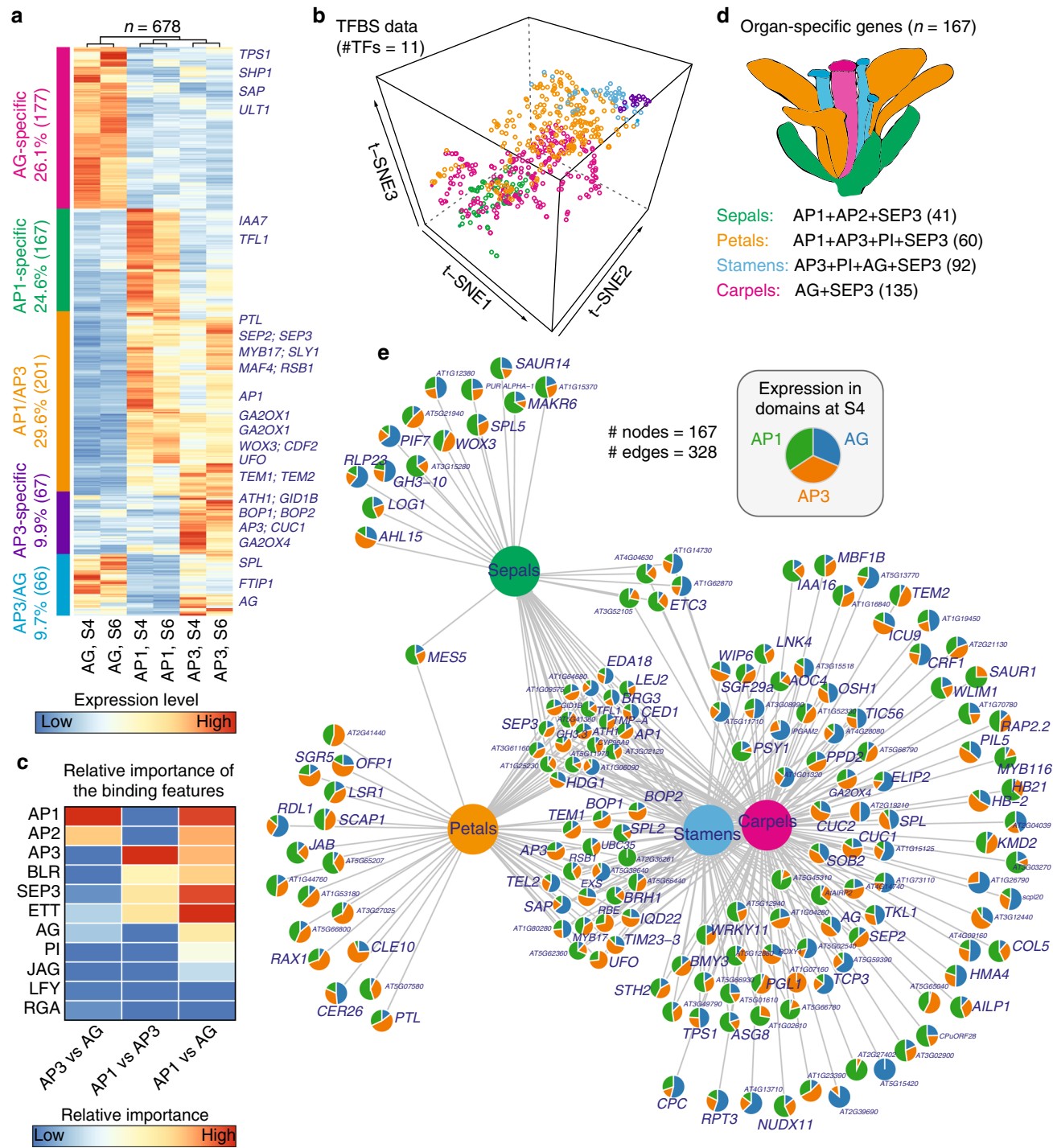

**Fig. 5** Organ-specific gene expression controlled by floral master regulators. **a** Clustering analysis of domain-specific gene expression. Five major clusters correspond to five distinct spatiotemporal domains (colored in left bars): AP1-, AP3-, AG-specific, "AP1/AP3" (AP1 and AP3-common domains), and "AP3/ AG" domains. Dendrograms above the heatmap illustrate the overall similarity of gene expression patterns. Example genes are indicated in the right. Domain expression data at stages 4 (S4) and 6 (S6) were obtained from ref. [62]. **b** 3D t-SNE (t-distributed stochastic neighbor embedding) plot showing the overall similarity of transcription factor (TF)-binding profiles of organ-specific genes. Normalized ChIP-seq intensity for the 11 selected TFs were used in the analysis. **c** Predicting the contributions of promoter binding by multiple TFs to gene expression difference between two different domains by a Lasso regression model. Regression coefficients are plotted in heatmap as relative importance of the binding features. **d** Domain-specific genes targeted by floral quartets that determine different floral organ identities, i.e., sepals, petals, stamens, and carpels, as colored in the diagram of floral organs. The number of organ-specific genes is shown in parentheses. **e** Network showing domain expression patterns of target genes for the four floral homeotic tetrameric MADS TF complexes as indicated in **d**. Nodes as pie charts show gene expression patterns in AP1, AP3, and AG domains at stage 4. Edges represent target genes by the corresponding floral quartet complexes

FFLs are typically found in miRNA-containing regulatory networks, where a master TF regulates both a target and its miRNA with opposing (i.e., coherent feed-forward) or allied effects (i.e., incoherent feed-forward) on the expression level of their common target gene[70,71], representing as an important strategy to confer molecular robustness and thereby to achieve developmental canalization[70]. In the floral GRN constructed in this study, these loops are involved in either stable down-regulation or precisely timed upregulation of TF activities (Supplementary Data 6). Examples of coherent FFLs include the stable downregulation of miR156-controlled SPL TF activities during flower developmental progression, and the induction of AP2-like TF activities linked with the repression of *MIR172* by AP1 and SEP3 (Supplementary Fig. 8). These findings are in line with previous results which showed that AP2 is involved in restricting *AG* activities to the inner floral whorls[16]. Interestingly, *AP2* paralogs, *TOE1* and *SNZ*, show distinct expression pattern when compared to the *AP2* gene in flowers. This is probably due to the fact that these *AP2* paralogs mainly play repressive role in floral transition[20,21] while *AP2* mainly functions during flower development. We also identified coherent FFLs that result in the upregulation of *MIR319*-regulated TCP4 activities during flower development. Previous research has shown that proper control of TCP4 levels is important for regulation of petal and stamen growth[37]. Our results based on genomic binding data in combination with genetic analyses (Fig. 4) suggest that binding by SEP3 to the *TCP4* promoter activates *TCP4* during later stages of development, while SEP3 binds to the *MIR319a* promoter leading to its downregulation in flowers. Notably, the suppression of SEP3 to *MIR319a* expression seems to mainly occur at early stages of flower development since in stage 10 flowers, expression of *MIR319a* is comparable with that in wild type. Furthermore, we observed that the binding level of SEP3 in the *TCP4* promoter is decreased, but *TCP4* expression is increased at stage 10 as compared to stage 4 in flowers. This indicates the existence of other factors regulating *TCP4* expression at later stages of flower development. Interestingly, *SEP3* expression in the rice protoplast-based luciferase assay was associated with increased *MIR319a* and *TCP4* promoter activity. This contrasts the observation that enhanced *SEP3* expression in developing flowers is coupled to lower *MIR319a* expression. This suggests that SEP3 requires cofactors that are not present in the (leaf-derived) protoplast system to mediate gene repression. *MIR319a* directly targets *TCP4* mRNA for degradation to promote cell proliferation[60]. Thus our data suggest that the SEP3-MIR319a-TCP4 module helps to trigger stage-specific activation of *TCP4* expression, which is critical for determining final petal size[37]. Further stage-and-organ-specific expression analyses are needed to test this hypothesis in the future. Thus, this FFL regulates the timing of petal expansion, and thereby the final petal size. In addition, the genomic binding data indicate that SEP3 may act together with the other members of the floral homeotic "petal" complex, including PI, AP3, and AP1 in regulating *TCP4* and *MIR319a* (Supplementary Fig. 10d).

A combined analysis of genome-wide binding data and floral domain-specific translatome data allowed us to predict genes that are directly regulated by floral homeotic complexes in a tissue-specific manner. We found that binding of non-MADS TFs can further improve the prediction of tissue-specific gene expression patterns. In particular, BLR and ETT/ARF3 were predicted to contribute to organ-specific gene regulation. The homeobox TF BLR was previously identified as a complex partner of AP1[59], and it is a multifunctional TF that controls patterns of oriented cell divisions in the inflorescence meristem[42,72], initiation of flower meristems, and repression of the AG-related genes *SHP1* and *2* in the developing gynoecium[73,74]. BLR was also found to be

involved in repression of *AG* in the outer floral whorls, especially in later developing flowers and at higher temperatures[75]. Combinatorial control of gene expression by homeotic TFs and BLR may therefore help to robustly restrict reproductive organ identities to the inner floral whorls. In line with this, we found that BLR mostly contributes to differentiate gene expression in perianth and reproductive whorls of the flower (Fig. 5c), suggesting that BLR controls perianth development not only via repression of *AG*, but also of via regulation of other genes involved in organ development. Loss-of-function of the auxin-responsive TF gene *ETT* display pleiotropic defects in flower development, including increase in perianth organ numbers, decrease in stamen number, and patterning defects in the gynoecium[24]. The results of our analysis suggest that ETT DNA-binding mainly contributes to distinguish AP1 domain from AG domain, and to a lesser extent from the AP3 domain. In line with this, ETT becomes expressed in stage 1 FMs and is later expressed in a complex pattern within petal, stamen, and carpel organ primordia[24]. ETT has not been identified as a complex partner of floral homeotic proteins; however, our data suggest that combinatorial binding of ETT and homeotic TFs helps to determine patterns of tissue-specific gene expression patterns in the flower.

In summary, systematic integration of DNA-binding data with genome-wide mRNA and miRNA expression data allowed us to determine key-regulatory properties in the GRN that controls early stages of flower development, including specification of floral whorls, and organ differentiation and growth. We also identified a novel FFL that controls petal growth downstream of the floral homeotic proteins, and assessed the roles of combinatorial gene regulation by multiple TFs in determining domain-specific gene expression patterns. Using these network-based approaches, we identified novel potential candidate genes that are responsible in determining the distinct morphologies and sizes of different types of floral organs. Future research will be required to systematically experimentally study the functions of specific regulatory interactions, e.g., using targeted mutagenesis of *cis*-regulatory regions and TFBSs.

## Methods

**Plant materials**. pAP1::AP1-GR *ap1 cal* plants were grown in a growth chamber under long day condition (16 h light, 8 h dark) with 120 umol/m²/s light intensity while the temperature changed from 22 °C during daytime to 20 °C in the night. Inflorescences were harvested from pAP1::AP1-GR *ap1 cal* plants after bolting (around 2 cm high) and at 2, 4, and 8 days after the first treatment by dexamethasone (daily treatment with 2 µM dexamethasone, 0.01% (v/v) ethanol, and 0.01% Silwet L-77) for both mRNA-seq and miRNA-seq analysis.

**RNA-seq and ChIP-seq data analysis**. Detailed methodology for RNA-seq and ChIP-seq data analysis are provided in Supplementary Notes 1 and 2.

**Expression analysis for miRNA and other genes**. To measure the expression of miRNAs, the stem loop RT-qPCR method described by Chen[76] was applied with minor modification. In detail, total RNA was extracted by Trizol (Thermo Fisher Scientific, USA) and RNA was eluted with DEPC-treated water plus 1 µl RNAse-Inhibitor (Epicentre, USA). Three hundred nanograms DNase-treated RNA was used for stem–loop transcription by combining with a stem–loop primer (100 pm), 10 mM dNTPs, and RNAase/DNase free water in a final volume of 13.5 µl. The reaction was heated for 5 min at 65 °C and then chilled on ice for 1 min. Super Script III enzyme (Thermo Fisher Scientific, USA) was used to perform reverse transcription. For detecting mRNA gene expression, total RNA was extracted with the GenUP™ Plant RNA Kit (Biozyme, USA) and cDNA synthesis was performed using M-MuLV Reverse Transcriptase (NEB, USA). Quantitative PCR was performed by using SsoAdvanced™ Universal SYBR® Green Supermix (Biorad, USA) using a CFX96 Touch™ Real-Time PCR Detection System. The primer sequences used here can be found in Supplementary Data 8.

**Dual-luciferase assay**. The full coding sequence of *SEP3* was isolated and placed after a CaMV35s promoter by KpnI and BamHI into a effector plasmid pGreenII0800 (ref. 77). A 4.0 kb ATG-upstream fragment of *TCP4* was isolated to serve as a wild-type *TCP4* promoter and a wild-type *MIR319a* promoter was identified

as previously described[37]. Overlapping PCR was applied to remove the SEP3 ChIP-seq peaks in the promoters. GACGCAACTGGGTAAGATGACAGCCAAAGCTT CCTCGTTTCC CAA and TTGGGAAACGAGGAAGCTTTGGCTGTCATCTTA CCCAGTTGCGTC were used to remove the TSS proximal SEP3-binding region and TACACTGCGGATACAAGTATGCCCCAATAACTACATTTTCTACG and CGTAGAAAATGTAGTTATTGGGGCATACTTGTATCCGCAGTGTA are the primers used to generate mutated *MIR319a* promoter. To eliminate the distal SEP3-binding region, a short version of *TCP4* promoter was amplified by using the primers of TCATCGGGTCGATTGGAAACAGAGG and TGGTAGAGCATATT CGTCGAGACG. Protoplasts were isolated and transformed as described previously[78]. In short, protoplasts were isolated by digesting rice sheath strips in digestion solution for 4 h. After filtering through Nylon mesh (35 μm), the protoplasts were collected and incubated in W5 solution (2 mM MES, pH 5.7, 154 mM NaCl, 5 mM KCl, 125 mM CaCl₂) at room temperature (25 °C) for 1 h. For transformation, 10 μl of plasmids (6 μg) was gently mixed with 100 μl of protoplasts and 110 μl of PEG-CaCl₂ solution (0.6 M Mannitol, 100 mM CaCl₂, and 40% PEG4000). Transformed protoplasts were then collected by centrifugation and were re-suspended for overnight incubation, after which protoplasts were collected for luciferase activity analysis. The luciferase activity was detected with the Dual-Luciferase Reporter Assay System E1960 (Promega, USA).

**Network analysis**. To assess the robustness of networks containing miRNAs, we analyzed and compared the network structure of two kinds of networks: the "meta-network" consisting of both TF-directed and miRNA-mediated regulation, and the network without miRNA regulation. In principle, more robust network shows a higher degree of tolerance against the removal of nodes. We used the swan_combinatory function in the NetSwan package in R to calculate the resistance of networks due to the node removal, using a cascading scenario to remove nodes in the decreasing order of their betweenness. The loss of connectivity along with the fraction of nodes removed was plotted (Fig. 3b).

Next, we systematically examined the enrichment of motifs in the meta-network. We focused on the analysis of autoregulation (1-node), feedback loop (2-node), and 3-node motifs, which have particular biological implications[51]. The 1- and 2-node network motifs were searched by custom code, while the enrichment of 3-node motifs was performed by the FANMOD algorithm[54]. The significance of motifs were determined by comparing the number of observed motifs with the number found in 1000 randomly shuffled networks, as implemented in FANMOD. To determine if two targets in an SIM have possible physical interactions, protein–protein interaction (PPI) data were obtained from ref. [79]. For each master regulator, the percentage of its targets with potential PPIs in the associated SIM motifs was calculated. As a control, the same number of genes as the investigated master regulator were sampled from the whole list of target genes for all the 15 master regulators, and a similar percentage value was calculated as above. As a result, two sets of percentage values ($n = 15$) were obtained and compared with a Student's *t*-test (Fig. 3f).

**Defining domain-specific genes**. AP1-, AP3-, and AG-domain translatome data (by TRAP-seq) at stages 4 (S4) and 6 (S6) were obtained from ref. [62]. Domain-specific expressed genes were determined by ANOVA with a domain effect *P*-value <0.05 and fold change >2 among the three domains at S4 or S6. In total, 6072 genes were identified at this step, 2311 (38.1%) of which were bound by more than one of 11 selected TFs (including AG, AP1, AP2, AP3, BLR, PI, SEP3, LFY, JAG, ETT, and RGA) and used for further analysis. A gene was defined as AP1-specific (sepal) if it was highly expressed (more than two-fold change) in the AP1 domain but not in the AP3 nor AG domains. A similar rule applies to AG-specific (carpel) or AP3-specific genes. AP1/AP3-common (petal) genes were defined if they showed high expression in both AP1 and AP3 domains but not in the AG domain. Similarly, AP3/AG-common (stamen) genes were defined. As a result, we identified 1013 organ-specific genes and 678 of them were bound by more than one of the 11 TFs.

**Modeling TF binding contribution and gene expression**. We adopted a regression-based model to relate gene expression dynamics (678 organ-specific genes as describe above) and TF binding (11 floral regulators), as described in previous studies. Specifically, we related the fold change (FC) of gene expression between two organs to a linear combination of the binding intensity of 11 individual TFs and fit a log-linear model:

$$\log_2 Y_i = \sum_{j=1}^{11} \beta_j x_{ij} + \varepsilon_i,$$

where $Y_i$ is the average FC value of gene $i$ in S4 and S6 between two compared organs. The TF-binding score $x_{ij}$ for TF $j$ at gene $i$ was defined the normalized peak score. TF binding data were normalized and mean centered before modeling. Lasso regression was performed using the glmnet package in R to calculate the direct contribution (i.e., the parameter $\beta$) of each TF to the expression change. The optimal model was chosen by five repeats of ten-fold cross-validation using the caret package in R. The above analysis was performed via the Shiny application HTPmod[80].

**Statistical analysis and data visualization**. If not specified, all statistical analyses and data visualization were done in R. t-SNE (t-distributed stochastic neighbor embedding) analysis was performed by the Rtsne library and the output was visualized by our HTPmod online tool[80] (https://www.epiplant.hu-berlin.de/shiny/app/HTPmod/). Hive plots were generated using the HiveR library. Network visualization was done using the igraph library.

**Code availability**. The ChIP-seq data analysis pipeline is adapted from https://github.com/PlantENCODE/plantGRNs. All other custom computer code is available from the corresponding authors upon reasonable request.

## Data availability

All data supporting the findings of this study are available in the Article or in Supplementary Information files. mRNA-seq and miRNA-seq data have been deposited with the NCBI Gene Expression Omnibus (GEO) under accession numbers GSE110539.

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

## Acknowledgements

We would like to thank Johanna Müschner for her assistance with RNA-seq, and Lei Wang from Huazhong Agricultural University for the help with the luciferase reporter gene experiments. We would like to thank the Center for Information Technology and Media Management (ZIM) at Potsdam University for providing high performance computing resources. K.K. wishes to thank the Alexander-von-Humboldt Foundation and the Federal Ministry of Education and Research for support.

## Author contributions

D.C. and K.K. conceived and initiated the project. K.K. supervised the study. D.C. designed the study and performed data analyses. L.-Y.F. contributed to data collection and analysis. W.Y. performed the experimental validations. D.C., W.Y., and K.K. wrote the manuscript. All authors read and approved the final version of the manuscript.

## Additional information

**Competing interests:** The authors declare no competing interests.

