## [Peer Review File · Nature Communications]

Reviewers' Comments:

Reviewer #1:

Remarks to the Author:

The manuscript entitled "Architecture of gene regulatory networks controlling flower development in *Arabidopsis thaliana*" by Chen et al performed an integrative analysis of high-throughput data yielded from ChIP-seq, mRNA-seq, miRNA-seq, TRAP-seq experiments, which were conducted on different tissues of *Arabidopsis* floral organ by PI's lab or other labs. I consider the analysis is overall a comprehensive one. Some novel results were shown and may be considered to be the progress in plant science. However, the lack of detailed description make me question about the methods employed and the legitimacy of some results.

My major concerns include:

1. The fingerprints of a TF's binding in the promoter regions of target genes do not necessarily mean the TF exerts a regulation on each of these targets. Did you count all genes that have one or more fingerprints as true targets? Were your target genes filtered using mRNA-seq or miRNA-seq data? However, the complication is that filtering out unexpressed or lowly expressed genes (e.g. CBF) may leave out the repressed target genes of a TF. How did you handle this in your analysis?
2. Does the width of each color bar in Fig 1b defines the time range in which a TF is expressed or the plant materials were harvested for a ChIP-seq experiment? In either cases, do you think the bias will be generated when you tried to identify co-occurrence or co-binding of multiple TFs in the promoters of target genes?
3. What are some criteria for 568 FFWs identified? Supplemental Table 5 just listed these FFWs without giving any criteria. Also, could you label the directions of regulation within each FFW (it is hard to know which one is X, Y, or Z) ?
4. FC is sometimes very misleading, especially when the denominator is small while the numerator is relatively large. For example, Gene i has expression levels in the control and treatment tissue are 0.25 and 2.5, while Gene j has expression levels in the control and treatment tissue are 500 and 1000, The FCs for Gene i and j are 10 and 1.5 times. In this case, the FC for Gene i is more likely a random error or an arbitrary value while the FC for Gene j is presumably more reliable. Given this, do you think your modeling may generate some misleading information?
5. Not sure how authors obtained leaf-specific genes (CUC1, CUC2, BOP1, BOP2 etc) when they tried to identify domain-specific gene expression patterns (AP1, AP3 and AG domains)? How were the three domains selected and based on what?
6. Since public data (ChIP-seq) were used, It's better to clarify what are novel discoveries after your analysis. For example, MIR319a and TCP4 were identified by Nag et al. Is the involvement of SEP3 in MIR319 and TCP interaction a new discovery? I suggest the authors to modify the manuscript thoroughly so that readers can easily distinguish what are new discoveries and what are prior knowledge.

Minor problems

7. Fig 1d, why the numbers (604, 2830, ...5564) of targets is more than the peak numbers as shown in Fig 1c?
8. Fig 1f, how co-binding and co-regulated were defined?

9. FFW is usually expressed as FFL (Feed-forward loop).

10. I would suggest the authors add a section of Conclusion to summarize their new results/findings.

11. When you talk about anything you did in your study, you should use past tense. But in Abstract, there are several sentences are in the present tense. The authors should correct these in Abstract and also in the main text if there are some.

Reviewer #2:

Remarks to the Author:

The present manuscript by Chen et al. aims to reconstruct regulatory networks orchestrating flower development in the model plant *Arabidopsis thaliana*. The authors compared different ChIP-seq datasets from some of the central players in the integration of different cues that regulate reproductive transition, floral meristem formation and the ones involved in the specification and development of the different floral organs. Authors complement that already available information with a transcriptome (both mRNA and microRNA) time course taking advantage of a previously described method for the synchronization of floral organ development. Those complementary approaches, allowed them to characterize a feed-forward regulatory loop in which SEP3 regulates petal development by binding TCP4 and its miRNA regulator miR319 promoters.

Although the paper is well written, executed and of interest, the authors should provide more confirmatory experiments. For instance, there are no independent confirmation by qRT-PCR or northern blot of the vast majority of changes in miRNA levels or some selected transcripts besides the sequencing data. To actually show that SEP3 has a direct effect on TCP4 and miR319 expression, authors should provide further evidence like, for instance, transient assays in tobacco in which both TCP4 and miR319 promoters drive luciferase reporters and show the observed opposite behavior in the presence of SEP3. Ideally, such approach should be complemented by repeating the same experiment using promoters mutated on the SEP3 binding sites within both promoters.

It would also be interesting the overlap between the different replicates from the RNA-seq experiments, for instance using PCA analysis. Likewise, authors should provide the same comparison for the different replicates from the ChIP-seq analysis.

Reviewer #3:

Remarks to the Author:

In this manuscript, the authors tried to construct a comprehensive gene regulatory network that governs the floral organ initiation and specification. For this purpose, they reanalyzed the ChIP-seq data of key floral transcription factors from published results, and combined them with the mRNA and miRNA expression profiles of the floral tissues at early stages in development. The authors particularly focused on the relations of the central floral transcription factors with miRNAs and their target genes, identified several modes of regulations in the miRNA-mediated gene network, and specifically studied the feed-forward loop in the three-node motif category using SEP3-miR319-TCP4/10 regulation as an example. They also tried to generate a floral organ specific regulatory network by integrating floral domain-specific transcriptome data with the ChIP-seq and RNA-seq results and identify key regulators and pathways in this network. Current studies in the field have uncovered the regulatory circuits of a number of floral transcription factors, but how these pathways function together to pattern the floral organ morphogenesis is still largely unknown. Results of this work will potentially contribute to our future research in this perspective. This manuscript contains an ample amount of bioinformatic data, however, the biological validation is a

bit weak. I think the manuscript requires major revision before publication.

My major concern lies in their analysis of the regulation of SEP3 on miR319 and TCP4/10 (Fig. 4). First of all, I am not quite convinced by their interpretation that the correlation between the SEP3 binding intensity and the expression levels of miR319 and TCP4 is associated with the timing of petal expansion (Fig. 4b and discussion). Petals develop much later than other floral organs. Petal cells are rapidly dividing during stage 7 and 8 and may start to expand and differentiate from stage 9 (Irish et al. 2009). So the stages assayed in Fig. 4 only include the phase of cell division during petal development. In addition, the function of TCP4 is to promote the differentiation of plant organs, and its expression persists in (actually is more restricted to) the petals at later stages (Sarvepalli et al 2001; Huang et al. 2016). Therefore, the decreased binding of SEP3 on TCP4 and the concomitant reduction of TCP4 expression (in the whole inflorescence) might be due to the dilution by other growing tissues or some other factors in their experiment. I think, in order to clarify the relation of the SEP3- miR319-TCP4 pathway with the petal developmental process, petal-specific gene expression and transcription factor binding assays need to be conducted and the tissues in the relevant growth stages need to be sampled in the experiments. In addition, the phenotypic analyses need to be done in a more detailed way. Although the whole flower images show the differences of the mutants and transgenic lines, single floral organs, particularly single petal images need to be presented (single petals of only three lines are shown in Suppl. Fig. 5 and the images are not in a good quality), and statistical analyses need to be done to show the differences in petal size and shape in these plants. I am also not quite convinced by their qRT-PCR results. Since the double mutant F2-6 has an intermediate phenotype, why does it have the lowest expression of MIR319a and highest expression of TCP4 among all the plants?

My second question comes from the characterization of regulators in the organ-specific networks. In line 263-267, the authors mentioned three categories of genes "sets of organ-specific target genes; commonly regulated genes that regulate leaf development; genes that need to be modulated in an organ-specific manner to generate the distinct morphologies of different types of floral organs.". Then they specifically pointed out CUC1/2, BOP1/2, SAP and PPD1/2, which is quite confusing. Are these genes belonging to any of the above categories? More explanations need to be written here and in the discussion to show how these genes are involved in the organ-specific regulatory network of flowers. Also, "PEAPOD1 (PPD2)" is PPD1 or 2? The regulation of SAP on PPD genes also needs a reference.

Other issues:

1. In the introduction, the authors might want to briefly mention the function of BLR, JAG, ETT and RGA in flower development since the downstream pathways of these genes are analyzed in the results.
2. In Fig. 4b, the first and third charts need labels on their X-axis.

Reviewer #4:

Remarks to the Author:

The authors used a combination of ChIP-seq, mRNA and miRNA-seq data to reconstruct the dynamic GRN controlling floral meristem development and organ differentiation. The idea is quite novel and the main results will contribute to the research in this field.

In the first section, the authors integrated a lot of ChIP-seq data to gain a comprehensive view of TF binding sites. These data were generated in different labs and under different conditions. So it might be important to provide some evidence to justify the difference between each individual ChIP-seq experiment and to make them comparable? It will be even better if the authors could provide some validated evidence about the reliability.

Combine the ChIP-seq data with the mRNA-seq, the authors identified some important genes during the reproductive phase. It will be nice if the authors could show systemic correlation between gene expression changes and the bound of TFs such as the positive or negative correlation. This will strengthen the conclusion generated from the data.

The authors investigated miRNA-mediated gene regulatory networks and enriched network motifs. The FFW loop motifs is the most interesting one. SEP3 regulating both MIR319 and TCP4 was used

as an example for the FFW loop. But the authors should claim clearly which data was generated in this work and which was from the previous study. For example, MIR319a mutant and overexpression was not cited in the text (line 219). The phenotype of the mutants has been described in original paper and the authors only used them to justify the conclusion. It will mislead the readers to overestimate the work in this paper.

In summary, the major claims in the paper will influence thinking in this field. It will lead some following research to study molecular networks underlying floral organ specification in plants. It should be suitable for publishing if the points mentioned above have been justified.

Reviewers' comments:

Reviewer #1 (Remarks to the Author):

The manuscript entitled "Architecture of gene regulatory networks controlling flower development in *Arabidopsis thaliana*" by Chen et al performed an integrative analysis of high-throughput data yielded from ChIP-seq, mRNA-seq, miRNA-seq, TRAP-seq experiments, which were conducted on different tissues of *Arabidopsis* floral organ by PI's lab or other labs. I consider the analysis is in overall a comprehensive one. Some novel results were shown and may be considered to be the progress in plant science. However, the lack of detailed description make me question about the methods employed and the legitimate of some results.

My major concerns include:

1. The fingerprints of a TF's binding in the promoter regions of target genes do not necessarily mean the TF exerts a regulation on each of these targets. Did you count all genes that have one or more fingerprints as true targets? Were your targets genes be filtered using mRNA-seq or miRNA-seq data? However, the complication is that filtering out unexpressed or lowly expressed genes (e.g. CBF) may leave out the repressed target genes of a TF. How did you handle this in your analysis?

Response: We thank the reviewer for the comments. We agree with the reviewer that TF binding doesn't necessarily mean regulation.

In the initial analysis, we indeed counted all genes that have at least one fingerprint as true targets (**Fig. 1d**). In the subsequent analyses, in order to describe the role of these genes in the gene regulatory networks, the target genes were further filtered using mRNA-seq (**Fig. 2c-e**) and miRNA-seq data (**Fig. 3; Supplementary Fig. 7 and Supplementary Data Set 1**). In fact, in all detailed analyses (**Figs. 2-4**), we mainly focused on the target genes that were differentially expressed ($FC > 2$ and $FDR < 0.05$), and we do not consider target genes without significant expression change during the investigated developmental stages.

In the RNA-seq data analysis, we filtered genes with a cutoff of FPKM > 0.1 (see **Methods**). We believe some lowly expressed or repressed genes are still in the list since the used cutoff is quite lenient (compared with a maximum FPKM=9735 and a median FPKM=14.2). However, in order to identify the true target genes for floral regulators, we did not consider lowly expressed genes without any significant change during flower development. Indeed, we found that DE genes were

significantly overlapped with floral regulator target genes (p-value $< 1.1 \times 10^{-91}$; hypergeometric test; **Fig. 2a**).

2. Does the width of each color bar in Fig 1b defines the time range in which a TF is expressed or the plant materials were harvested for a ChIP-seq experiment? In either cases, do you think the bias will be generated when you tried to identify co-occurrence or co-binding of multiple TFs in the promoters of target genes?

Response: We thank the reviewer for pointing this out. The width of each bar in **Fig. 1b** roughly defines the time range in which the plant materials were harvested for a TF ChIP-seq experiment. We have mentioned this in the figure legend in the revision.

There might be some bias to identify co-occupancy or co-binding of multiple TFs when we compare the data in different stages. However, ChIP-seq data generated in the same or similar developmental stages can be used for direct comparison, especially for the proteins that are involved in one heteromeric complex as we discussed in the main text: “*MADS proteins controlling floral transition (i.e. FLC, FLM and SVP) have similar binding patterns, suggesting that they may interact with each other to exert their functions in the flowering pathway*^{8,41,42}. Similarly, there is a high overlap of binding sites of the MADS proteins in the AG-SEP3-AP3-PI (stamen) heterotetramers (**Fig. 1f,g**).”. In fact, we did not mean to emphasize that these targets are simultaneously bound by all the mentioned master TFs. We would show that these targets could be bound by those TFs, probably in a developmental stage-specific manner.

3. What are some criteria for 568 FFWs identified ? Supplemental Table 5 just listed these FFWs without giving any criteria. Also, could you label the directions of regulation within each FFW (it is hard to know which one is X, Y, or Z) ?

Response: We thank the reviewer for pointing out this issue. The 568 FFLs were itemized from the ‘meta-network’ as presented in **Fig. 3a**, where only expressed miRNAs (by miRNA-seq) and differently expressed TFs (by mRNA-seq) were taken into consideration. In the revision, we further emphasized the AP1- and SEP3-regulated FFLs, for which the corresponding time-series gene expression data and TF binding data were both available (**Supplementary Figs. 7 and 8**). The expression change of a TF gene could be due to either the expression change of its miRNA regulator(s) or the binding change of its upstream master regulator(s). We found that 73.3%

(99/135) of the AP1- and SEP3-regulated FFLs were supported by significant gene expression changes as well as TF binding changes (**Supplementary Data Set 1**), and were thus considered as developmentally changing loops for functional investigation.

We also thank the reviewer for the nice suggestion. We have described the criteria used for FFL identification in the **Supplementary Table 5** in the revised version of the manuscript. We have also labeled the directions of regulation as well as the significant level for each FFL in the **Supplementary Data Set 1**.

4. FC is sometimes very misleading, especially when the denominator is small while the numerator is relatively large. For example, Gene i has expression levels in the control and treatment tissue are 0.25 and 2.5, while Gene j has expression levels in the control and treatment tissue are 500 and 1000, The FCs for Gene i and j are 10 and 1.5 times. In this case, the FC for Gene i is more likely a random error or an arbitrary value while the FC for Gene j is presumably more reliable. Given this, do you think your modeling may generate some misleading information?

Response: We thank the reviewer for this interesting comment. In our analysis, we only considered domain-specific expressed genes that were determined by ANOVA statistics with a domain effect p-value < 0.05 and fold change > 2 among the three domains at S4 or S6 (three replicates each). In this regard, gene i with lower average values doesn't mean that its expression looks more likely a random error, but instead, it indicates its true lower expressed status. So we believe that the fold change values are comparable between genes i and j. Therefore, the modeling analysis may not generate misleading information due to this issue.

5. Not sure how authors obtained leaf-specific genes (CUC1, CUC2, BOP1, BOP2 etc) when they tried to identify domain-specific gene expression patterns (AP1, AP3 and AG domains)? How were the three domains selected and based on what?

Response: We thank the reviewer for this useful remark and we are sorry for the confusion due to unclear description by the sentence. Here, we took *CUC1/2*, *BOP1/2* and *SAP/PPD* as examples of genes that play role in floral organ specification. Essentially, *CUC1/2* and *BOP1/2* have multiple roles in plant development, including floral patterning. We have now updated the corresponding sections in the manuscript, adding references to the relevant literature: “Using this approach, we were able to recover sets of organ-specific vs. common target genes, as well as

genes that need to be modulated in an organ-specific manner to generate the distinct morphologies of different types of floral organs. Those genes showing floral domain-specific expression patterns include CUP-SHAPED COTELYDON1 (CUC1) and CUC2⁶⁷, BLADE-ON-PETIOLE 1 (BOP1) and BOP2⁶⁸, as well as the F-box gene STERILE APETALA (SAP) and its target PEAPOD2 (PPD2)⁶⁹. CUC1 and CUC2 are redundantly acting factors required for establishing boundaries of floral primordia^{70,71}. Similarly, BOP1 and BOP2 are expressed in the boundaries of floral primordia, and they regulate FM identity⁷². Both CUC1/2 and BOP1/2 genes are preferentially expressed in the AP3 domain, whereas SAP and PEAPOD2 (PPD2) showed elevated expression in the AG domain (Fig. 5e). SAP and PPD2 are known to regulate flower organ size and fertility through an E3 ubiquitin ligase mediated protein degradation pathway⁶⁹.”

The AP1, AP3 and AG domains were selected based on domain-specific translome data (Jiao et al. 2010). See more details about “*Defining domain-specific genes*” in the Methods section.

6. Since public data (ChIP-seq) were used, It's better to clarify what are novel discoveries after your analysis. For example, MIR319a and TCP4 were identified by Nag et al. Is the involvement of SEP3 in MIR319 and TCP interaction a new discovery? I suggest the authors to modify the manuscript thoroughly so that readers can easily distinguish what are new discoveries and what are prior knowledge.

Response: We thank the reviewer for the comment. Indeed, the regulation of petal development by *MIR319a*-TCP module has been reported. In our study, we described the petal phenotype of *sep3* mutant as well as SEP3 enhanced expression line and compared that with *MIR319a* mutant phenotype. We also showed further genetic and molecular evidence to support the finding that SEP3 is upstream regulator of *MIR319a* and *TCP4*. These three factors form a FFL loop to regulate petal development. Involvement of SEP3 in *MIR319a* and *TCP4* interaction is a new discovery in our work. Following the reviewer’s suggestion, we modified the manuscript to focus on the novel aspects and discuss them in the context of prior knowledge (in the section of “*SEP3 as a Direct Regulator of MIR319a in Controlling Petal Growth*” in pages 8 and 9).

Minor problems

7. Fig 1d, why the numbers (604, 2830, ...5564) of targets is more than the peak numbers as shown in Fig 1c?

Response: Thank you for the comment. This is due to the principle of target assignment we used (“For each TF, target genes were assigned when their transcription start sites (TSSs) are positioned within 3 kb from the peak regions. If multiple peaks can be assigned to a same gene, only the closest peak was considered as its putative binding site”). Some peaks can be assigned to multiple target genes.

8. Fig 1f, how co-binding and co-regulated were defined?

Response: We thank the reviewer for pointing this out. In fact, we had defined this in the figure legend. To make this clearer, we have improved the description in this revision: *“Upper triangle, highly significant co-regulation relationships are assigned a dark blue color, representing $-\log_{10}(p\text{-value})$. The p-value was calculated from a hypergeometric test to check the significance of target overlapping between any two TFs using all the annotated genes as background. Lower triangle, highly significant co-binding relationships are assigned a dark orange color. The significance of co-binding by any two TFs were tested by Jaccard statistics, which measures the ratio of the number of intersecting base pairs occupied by both TFs to the number of base pairs in their union”*.

9. FFW is usually expressed as FFL (Feed-forward loop).

Response: Agree. We have changed FFW as FFL throughout the manuscript. Thank you.

10. I would suggest the authors add a section of Conclusion to summarize their new results/findings.

Response: We thank the reviewer for this suggestion. We have added a Conclusion section after the Discussion part.

11. When you talk about anything you did in your study, you should use past tense. But in Abstract, there are several sentences are in the present tense. The authors should correct these in Abstract and also in the main text if there are some.

Response: We thank the reviewer for this suggestion. We have corrected this in the revision.

Reviewer #2 (Remarks to the Author):

The present manuscript by Chen et al. aims to reconstruct regulatory networks orchestrating flower development in the model plant *Arabidopsis thaliana*. The authors compared different ChIP-seq datasets from some of the central players in the integration of different cues that regulate reproductive transition, floral meristem formation and the ones involved in the specification and development of the different floral organs. Authors complement that already available information with a transcriptome (both mRNA and microRNA) time course taking advantage of a previously described method for the synchronization of floral organ development. Those complementary approaches, allowed them to characterize a feed-forward regulatory loop in which SEP3 regulates petal development by binding TCP4 and its miRNA regulator miR319 promoters.

Although the paper is well written, executed and of interest, the authors should provide more confirmatory experiments. For instance, there are no independent confirmation by qRT-PCR or northern blot of the vast majority of changes in miRNA levels or some selected transcripts besides the sequencing data.

Response: Thank you for the suggestions. We now examined the dynamic expression of several miRNAs (miRNA319a, miRNA156c, miRNA172d, miRNA157d and miRNA167a) by stem-loop RT-qPCR and the expression of some function known TFs (like AP2, TOE1 and SNZ) by qRT-PCR during flower development. In general, the experimental data agree well with the NGS data (overall Pearson's correlation $r = 0.51$). We showed these data in the **Supplementary Fig. 8**.

Supplementary Fig. 8: Validation of miRNA-seq and RNA-seq data. (a) Validation of expression levels during flower development for selected genes, either by stem-loop RT-qPCR (for miRNA genes) or by qRT-PCR (for TFs). Tissues representing different developmental stages flowers were harvested from *pAPI::API-GR ap1cal* plants according days of DEX induction (DAI) for expression analysis. U6 and *Tip41* were taken as internal control to normalize the expression level of miRNA genes and other transcription factors, respectively. Result shown here represents data from three independent biological replicates. (b) Correlation of gene expression levels by experimental validation and sequencing results. For visualization purpose, expression data for each gene were centered to the mean value and scaled as *s.t.d.*=1. Pearson correlation coefficients (*r*) between experimental data and sequencing data for the selected genes were shown for each stage. Bar=*mean*±*s.d.*.

To actually show that SEP3 has a direct effect on TCP4 and miR319 expression, authors should provide further evidence like, for instance, transient assays in tobacco in which both TCP4 and

miR319 promoters drive luciferase reporters and show the observed opposite behavior in the presence of SEP3. Ideally, such approach should be complemented by repeating the same experiment using promoters mutated on the SEP3 binding sites within both promoters.

Response: Thank you for the nice suggestion. Following the suggestion from the reviewer, we performed a dual luciferase reporter-based transient expression assays to detect the direct effect of SEP3 binding on promoters of *MIR319a* and *TCP4*. We included both wild-type promoters and promoters in which the binding motif of SEP3 were removed. We show the new experimental data in the **Supplementary Fig. 10**.

We observed that adding of SEP3 resulted increased luciferase activity either when luciferase gene is driven *MIR319a* promoter or by *TCP4* promoter, which is contradictory to the observation that enhanced *SEP3* expression leads to lower *MIR319a* expression. But this is not surprising since *SEP3* has gene activation activity, especially considering the fact that the promoters use in this system are always accessible and the SEP3 complex partners are different in flower and other plant tissues. However, the results showed that deletion of both SEP3 binding peak in *MIR319a* promoter and far-upstream but not the TSS proximal in *TCP4* promoter could significantly lower the effect of SEP3 on luciferase expression. Further *in vivo* evidence, such as inducing *SEP3* expression in various genetic backgrounds with modified *MIR319a* or *TCP4* promoters, will be needed to get a deeper understanding of (stage-specific) SEP3 action on *MIR319a* and *TCP4* promoters.

Supplementary Fig. 10: Effects of SEP3 direct binding to TCP4 and MIR319a promoters. (a) Diagrams showing the reporter and effector constructs of dual-luciferase reporter based transient assay. CaMV35S promoter driven SEP3 is the effector and different versions of TCP4 or MIR319a promoters (with or without a SEP3 site peak (BS)) driven firefly luciferase (LUC) are served as reporters. CaMV35S promoter driven renilla luciferase (REN) was simultaneously expressed with test promoters in the same reporter construct to served as internal control. (b) Relative LUC to REN ratio of co-transfection of SEP3 and different reporters. The ratio of LUC/REN of the empty vector plus corresponding promoter (CaMV35S promoter alone as effector; CaMV35S promoter driven REN together with test promoter driven LUC as reporter) was considered as a calibrator (set as 1). Data represents the mean \pm s.e. of five independent replicates. Significance codes, *: p -value $<$ 0.05 and NS: not significant, by Student's t -tests.

It would also be interesting the overlap between the different replicates from the RNA-seq experiments, for instance using PCA analysis. Likewise, authors should provide the same comparison for the different replicates from the ChIP-seq analysis.

Response: We thank the reviewer for this comment. Following the suggestion by the reviewer, we have performed PCA analysis on the RNA-seq data (Supplementary Fig. 5c) as well as the ChIP-seq data (Supplementary Fig. 2).

Supplementary Fig. 5c. Principal component analysis (PCA) of RNA-seq experiments (as labeled in different colors) using all the expressed protein-coding genes, as measured in FPKM (fragments per kilobase of transcript per million mapped reads). Percentage values in parentheses indicate percentage of variance explained by each PC.

Supplementary Fig. 2. Assessment of public ChIP-seq experiments. Principal component analysis (PCA) of ChIP-seq profiles for the investigated transcription factors (TFs) in this study using peak scores of all the target genes. Peak scores were generated by MACS2. Missing values were assigned to 1 (the lowest peak score). The top three principle components (PCs) were shown. Percentage values in parentheses indicate percentage of variance explained by each PC.

Reviewer #3 (Remarks to the Author):

In this manuscript, the authors tried to construct a comprehensive gene regulatory network that governs the floral organ initiation and specification. For this purpose, they reanalyzed the ChIP-seq data of key floral transcription factors from published results, and combined them with the mRNA and miRNA expression profiles of the floral tissues at early stages in development. The authors particularly focused on the relations of the central floral transcription factors with miRNAs and their target genes, identified several modes of regulations in the miRNA-mediated

gene network, and specifically studied the feed-forward loop in the three-node motif category using SEP3-miR319-TCP4/10 regulation as an example. They also tried to generate a floral organ specific regulatory network by integrating floral domain-specific transcriptome data with the ChIP-seq and RNA-seq results and identify key regulators and pathways in this network. Current studies in the field have uncovered the regulatory circuits of a number of floral transcription factors, but how these pathways function together to pattern the floral organ morphogenesis is still largely unknown. Results of this work will potentially contribute to our future research in this perspective. This manuscript contains an ample amount of bioinformatic data, however, the biological validation is a bit weak. I think the manuscript requires major revision before publication.

My major concern lies in their analysis of the regulation of SEP3 on miR319 and TCP4/10 (Fig. 4). First of all, I am not quite convinced by their interpretation that the correlation between the SEP3 binding intensity and the expression levels of miR319 and TCP4 is associated with the timing of petal expansion (Fig. 4b and discussion). Petals develop much later than other floral organs. Petal cells are rapidly dividing during stage 7 and 8 and may start to expand and differentiate from stage 9 (Irish et al. 2009). So the stages assayed in Fig. 4 only include the phase of cell division during petal development. In addition, the function of TCP4 is to promote the differentiation of plant organs, and its expression persists in (actually is more restricted to) the petals at later stages (Sarvepalli et al 2001; Huang et al. 2016). Therefore, the decreased binding of SEP3 on TCP4 and the concomitant reduction of TCP4 expression (in the whole inflorescence) might be due to the dilution by other growing tissues or some other factors in their experiment. I think, in order to clarify the relation of the SEP3-miR319-TCP4 pathway with the petal developmental process, *petal-specific gene expression* and *transcription factor binding assays* need to be conducted and the tissues in the relevant growth stages need to be sampled in the experiments.

Response: Thank you for these useful comments. We performed new experiments by following the suggestions of the reviewer. Firstly, using pAP1::AP1-GR *apl cal* inducible lines, we detected the binding of SEP3 to *TCP4* in stage 10 flowers. SEP3 showed decreased occupancy to *TCP4* promoters at stage 10 compared with stage 4. We also compared the expression of *TCP4* at these two stages and the results showed that *TCP4* is highly expressed at stage 10, which supports the fact that *TCP4* mainly expressed in later stages of flower development mentioned by the reviewer (**Fig. R1** below). Secondly, we checked the expression of *TCP4* and *MIR319a* in stage 10 flowers (it was not possible to dissect stage 10 petals) of both enhanced *SEP3* expression line

and wild-type plants. Interestingly, we observed clear elevated expression of *TCP4* in *SEP3* enhanced expression lines but the expression of *MIR319a* showed no difference between these two genotypes although the difference in expression of *MIR319a* persists while using the whole inflorescences (Fig. 4e). Based on these data, we agree with the reviewer that our previous interpretation of *SEP3* binding intensity and the expression levels of *MIR319a* and *TCP4* is associated with the timing of petal expansion is inappropriate, and we changed the manuscript accordingly. Considering the hypothesis that miR319a directly targets *TCP4* mRNA for degradation to promote cell proliferation (Huang and Irish, 2016), it might be that the *SEP3*-*MIR319a*-*TCP4* module helps to establish stage-specific *TCP4* expression levels, since this is critical for proper petal development. Further stage-and-organ-specific expression analyses are needed to test this hypothesis in the future. Due to technical limitations, we were not able to harvest specific petal tissue from stage 10 flowers. Therefore we used stage 10 flowers to perform the expression analyses.

Fig. R1. The binding level of *SEP3* to two genomic regions in *TCP4* promoters and relative expression of *SEP3* and *TCP4*. Enrichment of *SEP3* was detected by ChIP-qPCR using stage4 and stage10 flowers of *pAPI::API-GR ap1cal* plants. ChIP was performed using tissues from three independent harvesting. For each biological replicate, two technical repeats were performed. Expression of *TCP4* and *SEP3* were detected using the same tissues that were used for ChIP-qPCR. The relative expression represents data from three biological replicates.

Fig. 4e. Relative expression of *TCP4* and *MIR319a* in either whole inflorescences or stage 10 flower of WT and *SEP3* enhanced expression lines. For all the expression analysis, data represents mean of three independent biological replicates. *Tip41* gene (*AT4G34270*) was used as reference. Significance codes, *: p -value < 0.05, and NS: not significant, by Mann-Whitney tests. Bar=mean \pm s.e..

In addition, the phenotypic analyses need to be done in a more detailed way. Although the whole flower images show the differences of the mutants and transgenic lines, single floral organs, particularly single petal images need to be presented (single petals of only three lines are shown in Suppl. Fig. 5 and the images are not in a good quality), and statistical analyses need to be done to show the differences in petal size and shape in these plants.

Response: Thank you for the suggestions. We re-made all the flower related pictures by dissecting all the organs of a flower, especially petals and quantified the size of petal as well as other organs of an open flowers. Results for statistical analysis have been shown in **Fig. b,c** and **Supplementary Fig. 9** in the revised version of the manuscript.

Fig. 4b-c. (b) Pictures of single flowers and separated petals as well as other flower organs (top panel) and inflorescences (bottom) of wild-type (WT) plants, enhanced expression of *SEP3* (*SEP3^{HE4.1k}*), *jaw-D* mutant lines, and combination (F2) of *jaw-D* mutant and *SEP3^{HE4.1k}*. (c) Boxplots showing the petal width (left) and petal length (right) of different genotypes in b. Significance codes, ****: p -value < 0.0001, *: p -value < 0.05, and NS: not significant, by Mann-Whitney tests. Bar=mean \pm s.e..

Supplementary Fig. 9. *SEP3* and *MIR319a* have opposite role in petal development. (a) Pictures of single flower and separated flower organs of wild type (WT), *sep3-2* mutant and *MIR319a* overexpression line (*jaw-D*). (b) Performance of single flower and each whorl of flower organs of WT, *SEP3* enhanced expression line (*SEP3^{HE4.1k}*) and *MIR319a* mutant line (*miR319a¹²⁹*). (c) Boxplots showing the petal width (left) and petal length (right) of WT, *sep3-2*, *jaw-D*, *SEP3^{HE4.1k}* and *miR319a¹²⁹* plants. Flowers from five to eight plants with the same genotype were used for data collection. Five flowers were harvested from the main inflorescences of one single plant. Data represents the mean \pm s.e.. Significance codes, ****: *p*-value < 0.0001, *: *p*-value < 0.05, and NS: not significant, by Mann-Whitney tests.

I am also not quite convinced by their qRT-PCR results. Since the double mutant F2-6 has an intermediate phenotype, why does it have the lowest expression of *MIR319a* and highest expression of *TCP4* among all the plants?

Response: We thank the reviewer for pointing out this issue. Indeed, it is surprising that F2-6 has even lower *MIR319a* expression than that in *SEP3* enhanced expression lines. We re-checked the genotype and the expression of *MIR319a* and *TCP4* in F2-6 plants. The genotype is right and this line again gave low *MIR319a* expression. Since this F2-6 is the seeds from a cross did 8 years ago in Wageningen and is the only seed left from that cross, we failed to find out the reason why this line showed so low *MIR319a* expression. However, in order to get solid results, we used three newly generated F2 lines to repeat the experiment and F2-6 line was taken as control. The result showed that all the three new lines possess comparable miRNA319a expression with wild type but only F2-6 displayed again very low *MIR319a* and highest *TCP4* expression (**Fig. R2** below). We will use the data from the three new lines (see **Fig. 4d** in the revision). The phenotype data have been also collected using these new lines.

Fig. R2. Relative expression of *MIR319a*, *TCP4* and *TCP10* in the inflorescences of WT, *jaw-D* mutant, *SEP3*^{HE4.1k} lines and three independent F2 lines (F2-1/2/3) produced by crossing *jaw-D* and *SEP3*^{HE4.1k} lines. F2-6 was the old genotype generated in Wageningen.

My second question comes from the characterization of regulators in the organ-specific networks. In line 263-267, the authors mentioned three categories of genes” sets of organ-specific target genes; commonly regulated genes that regulate leaf development; genes that need to be modulated in an organ-specific manner to generate the distinct morphologies of different types of floral organs.”. Then they specifically pointed out CUC1/2, BOP1/2, SAP and PPD1/2, which is quite confusing. Are these genes belonging to any of the above categories? More explanations need to be written here and in the discussion to show how these genes are involved in the organ-

specific regulatory network of flowers. Also, “PEAPOD1 (PPD2)” is PPD1 or 2? The regulation of SAP on PPD genes also needs a reference.

Response: Thank you for the comments. We are sorry for the confusion and the mistake. *CUC1/2*, *BOP1/2*, *SAP* and *PPD1/2* are the genes that play role in floral organ differentiation. “*PEAPOD1*” should be *PEAPOD2*. Following the suggestions of the reviewer, we have specified the gene identities and described in more details how these genes affect flower development by saying “*Those genes showing floral domain-specific expression patterns include CUP-SHAPED COTELYDON1 (CUC1) and CUC2*⁶⁷, *BLADE-ON-PETIOLE 1 (BOP1) and BOP2*⁶⁸, as well as the *F-box gene STERILE APETALA (SAP) and its target PEAPOD2 (PPD2)*⁶⁹. *CUC1 and CUC2 are redundantly acting factors required for establishing boundaries of floral primordia*^{70,71}. Similarly, *BOP1 and BOP2 are expressed in the boundaries of floral primordia, and they regulate FM identity*⁷². Both *CUC1/2 and BOP1/2 genes are preferentially expressed in the AP3 domain, whereas SAP and PEAPOD2 (PPD2) showed elevated expression in the AG domain (Fig. 5e)*. *SAP and PPD2 are known to regulate flower organ size and fertility through an E3 ubiquitin ligase mediated protein degradation pathway*⁶⁹”.

Other issues:

1. In the introduction, the authors might want to briefly mention the function of BLR, JAG, ETT and RGA in flower development since the downstream pathways of these genes are analyzed in the results.

Response: We thank the reviewer for this nice suggestion. We added a sentence to briefly introduce the functional role of *BLR*, *ETT*, *JAG* and *RGA* in controlling the growth of floral organs: “*Floral organ growth and differentiation is further controlled by other types of TFs, including BELLRINGER (BLR)*²⁰, *JAGGED (JAG)*^{21,22}, *ETTIN (ETT)*^{23,24} and *REPRESSOR OF GIBBERELLIC ACID (RGA)*²⁵. *BLR encodes a homeodomain TF and plays a central role in shoot apical meristem development by affecting meristem activity, phyllotaxis and floral organ patterning*²⁰. *JAG encodes a zinc finger TF that regulates organ boundary formation and growth*²⁶. *ETT is an auxin-responsive TF*²⁷ that regulates several aspects of flower morphogenesis^{23,24} and *RGA controls inflorescence development via participating gibberellin signalling pathway*^{25,28}.”

2. In Fig. 4b, the first and third charts need labels on their X-axis.

Response: Thank you for the suggestion. We have added labels on the x-axis in charts of old Fig. 4b. Due to re-organization of the manuscript, this figure part has been moved the Supplementary Data Set 1.

Reviewer #4 (Remarks to the Author):

The authors used a combination of ChIP-seq, mRNA and miRNA-seq data to reconstruct the dynamic GRN controlling floral meristem development and organ differentiation. The idea is quite novel and the main results will contribute to the research in this field.

In the first section, the authors integrated a lot of ChIP-seq data to gain a comprehensive view of TF binding sites. These data were generated in different labs and under different conditions. So it might be important to provide some evidence to justify the difference between each individual ChIP-seq experiment and to make them comparable? It will be even better if the authors could provide some validated evidence about the reliability.

Response: We thank the reviewer for the comment about the reliability. The reviewer is right that the ChIP-seq data used in this study were generated in different labs and under different conditions. To make results from different ChIP-seq experiment more comparable, we adopted the IDR (Irreproducible Discovery Rate) framework (<https://sites.google.com/site/anshulkundaje/projects/idr/>) as a unified approach to derive highly stable thresholds based on reproducibility. In another word, the IDR method allows to identify true TF-regulated genes by separating high confidence/enrichment (signal) from low confidence/enrichment (noise) in the analysis. Indeed, we found that our reanalyses generally outperform the correspondingly original studies in terms of identifying true target genes by integration of differential expression data (when available) upon perturbation of the corresponding TF (**Supplementary Fig. 3b**). Furthermore, PCA of ChIP-seq profiles showed that ChIP-seq replicates and TFs with similar functions tend to group together (**Supplementary Fig. 2**). Furthermore, clustering analysis revealed that the regulators which are known to function as protein complex partners (e.g., FLC/FLM/ SVP, or AG/SEP3/AP3/PI) show very similarity of binding profile (**Fig. 1f**), even though these ChIP-seq experiments were generated in different labs.

Supplementary Fig. 2. Assessment of public ChIP-seq experiments. Principal component analysis (PCA) of ChIP-seq profiles for the investigated transcription factors (TFs) in this study using peak scores of all the target genes. Peak scores were generated by MACS2. Missing values were assigned to 1 (the lowest peak score). The top three principle components (PCs) were shown. Percentage values in parentheses indicate percentage of variance explained by each PC.

Supplementary Fig. 3b. Compare the number of target genes identified in this study (red), identified in the original studies (green), and differentially expressed genes identified in the corresponding mutant (blue). P-values are calculated by hypergeometric tests to test the significance of overlapping between TF regulated genes and differentially expressed genes.

Fig. 1f. Co-binding relationships and co-regulated targets by TFs. Upper triangle, highly significant co-regulation relationships are assigned a dark blue color, representing $-\log_{10}(p\text{-value})$. The p -value was calculated from a hypergeometric test to check the significance of target overlapping between any two TFs using all the annotated genes as background. Lower triangle, highly significant co-binding relationships are assigned a dark orange color. The significance of co-binding by any two TFs were tested by Jaccard statistics, which measures the ratio of the number of intersecting base pairs occupied by both TFs to the number of base pairs in their union. Dashed boxes indicate highly interplayed regulators in vegetative development (green) and flower development (purple).

Combine the ChIP-seq data with the mRNA-seq, the authors identified some important genes during the reproductive phase. It will be nice if the authors could show systemic correlation between gene expression changes and the bound of TFs such as the positive or negative correlation. This will strengthen the conclusion generated from the data.

Response: We thank the reviewer for this nice suggestion. We have performed correlation analysis between gene expression changes and the binding intensity of different TFs. The results are now shown in **Supplementary Fig. 6**.

Supplementary Fig. 6. Correlation of gene expression change and the binding intensity of different transcription factors (TFs). For each TF, contour plot showing the relationship between the TF binding intensity (in terms of peak score from MACS2) and the expression changes of its target genes. Expression change was measured as the difference of gene expression levels (in terms of FPKM) between stage 8 (S8) and S2. For the scatter plots, genes were binned into percentiles ($n=20$) based on their expression level, and the median expression and median binding intensity of each bin were plotted. Spearman's rank correlation ρ was shown.

The authors investigated miRNA-mediated gene regulatory networks and enriched network motifs. The FFW loop motifs is the most interesting one. SEP3 regulating both MIR319 and TCP4 was used as an example for the FFW loop. But the authors should claim clearly which data was generated in this work and which was from the previous study. For example, MIR319a mutant and overexpression was not cited in the text (line 219). The phenotype of the mutants has been described in original paper and the authors only used them to justify the conclusion. It will mislead the readers to overestimate the work in this paper.

Response: We thank the reviewer for pointing this out. Following the suggestion by the reviewer, we rewrote the SEP3-MIRNA319a-TCP4 part by firstly giving an introduction on what had been reported, including the petal phenotype of miRNA319a overexpression and mutant line as well as the effect of miRNA319a targeting *TCP4* on petal development. Then we described the results of our experimental analyses.

In summary, the major claims in the paper will influence thinking in this field. It will lead some following research to study molecular networks underlying floral organ specification in plants. It should be suitable for publishing if the points mentioned above have been justified.

Response: We thank the reviewer for the helpful comments and suggestions. We have revised the manuscript following the comments/suggestions by the reviewer, and we hope the revision is acceptable for publication.

Reviewers' Comments:

Reviewer #1:

Remarks to the Author:

I have reviewed the revised manuscript and think it is better than the previously version. All my previous comments were addressed and I do not have additional comments except a few minor ones.

1. You mentioned the Supplementary Fig. 2a, 2b in the main text, but your figure did not have a, and b labels in both figure and legends
2. Some complex Figures have a, b, c etc. labels etc. But they (a, b and c) were not cited in the main text.
3. Some new results, for instance PCA, were poorly explained. I do think you could add some explanation to your new results in the main text.

These need to be fixed.

Reviewer #3:

Remarks to the Author:

In the revised manuscript, the authors answered most of my questions. However, I think they still need to clarify the regulation of SEP3 on miR319 and TCP4 a bit more. Their new results indicate that SEP3 probably does not control later-stage flower development by regulating the miR319-TCP4 pathway. Although TCP4 expression is increased in SEP3[>], the occupancy of SEP3 on the TCP4 promoter is decreased and the expression of TCP4 is increased at stage 10 as compared to stage 4, which is inconsistent with their model that SEP3 positively regulates TCP4. In addition, the expression of MIR319a is not changed in SEP3[>] as compared to wild type at stage 10. So it is likely that the SEP3-miR319-TCP4 pathway only acts at early stages to control flower development. The authors may want to discuss these results more in detail in the manuscript.

Reviewer #4:

Remarks to the Author:

Feedback from Reviewer #4

For the reliability assessment of ChIP-seq data, the authors used IDR and PCA analysis indicating that the data from different conditions are comparable. The question has been well addressed. The authors also did the correlation analysis between gene expression changes and the binding intensity of different TFs. But the analysis of multiple target genes might generate a mixed result. The authors could include some examples of specific target genes to show the correlation which might be clearer.

The changes of the manuscript in regards to the previous work is acceptable.

In summary, I am happy for the paper to be published in NC after the minor points are addressed.

Feedback to the points addressed by Reviewer #2

The authors used stem-loop RT-qPCR to validate the expression of several miRNAs and qRT-PCR to validate the expression of some known TFs. The experimental data agree well with the NGS data. It is better to include the statistical analysis addressing the significance.

The luciferase reporter-based transient expression assays provided a strong evidence of a direct effect of SEPs binding on promoters of MIR319a and TCP4.

The PCA analysis also indicate a good overlap between different replicates from RNA-seq and ChIP-seq experiments.

The questions from reviewer #2 have been satisfactorily addressed. Minor points need be addressed further.

We thank the reviewers for their further comments on our manuscript. We have provided our point-by-point responses to reviewer comments. Our responses are marked in blue.

Reviewer #1 (Remarks to the Author):

I have reviewed the revised manuscript and think it is better than the previously version. All my previous comments were addressed and I do not have additional comments except a few minor ones.

1. You mentioned the Supplementary Fig. 2a, 2b in the main text, but your figure did not have a, and b labels in both figure and legends

Response: We thank the reviewer for pointing out this. We corrected this mistake in the revision.

2. Some complex Figures have a, b, c etc. labels etc. But they (a, b and c) were not cited in the main text.

Response: We thank the reviewer for noticing this. In this revision, we have tried to cite every figure panel when possible, including Supplementary Figures 4, 5 and 9.

3. Some new results, for instance PCA, were poorly explained. I do think you could add some explanation to your new results in the main text.

Response: Thanks for this comment. We have added explanation about the new analysis in the revision (on page 4): *"In this regard, the IDR (Irreproducible Discovery Rate) framework³⁴ was adopted as a unified approach to derive highly stable thresholds based on reproducibility. Principal component analysis (PCA) of ChIP-seq signal profiles revealed that datasets for ChIP-seq experiment replicates or for TFs with similar functions tend to group together (Supplementary Fig. 2)."*

Reviewer #3 (Remarks to the Author):

In the revised manuscript, the authors answered most of my questions. However, I think they still need to clarify the regulation of SEP3 on miR319 and TCP4 a bit more. Their new results indicate that SEP3 probably does not control later-stage flower development by regulating the miR319-TCP4 pathway. Although TCP4 expression is increased in SEP3[>], the occupancy of SEP3 on the TCP4 promoter is decreased and the expression of TCP4 is increased at stage 10 as compared to stage 4, which is inconsistent with their model that SEP3 positively regulates TCP4. In addition, the expression of MIR319a is not changed in SEP3[>] as compared to wild type at stage 10. So it is likely that the SEP3-miR319-TCP4 pathway only acts at early stages to control

flower development. The authors may want to discuss these results more in detail in the manuscript.

Response: Thank you for this insightful comment. Following the suggestion of the reviewer, we have added the following sentences in the Discussion: *“Notably, the suppression of SEP3 to MIR319a expression seems to mainly occur at early stage of flower development since in stage 10 flowers, expression of MIR319a is comparable with that in wild type. Furthermore, we observed that the binding level of SEP3 in the TCP4 promoter is decreased, but TCP4 expression is increased at stage 10 as compared to stage 4 in flowers, which indicates the existence of other factors regulating TCP4 expression at later stages of flower development.”*

<<see attached document for additional comments regarding response to reviewer #2>>

The authors answered most of the reviewer’s questions. There is only one concern that in the Supplementary Fig. 8, could the authors tell us why they chose these primary miRNA genes and miRNA target genes for the validation of miRNA-seq and RNA-seq results? In these genes, miR172 and AP2, SNZ and TOE1 are pairs of miRNA and its targets, but the target genes of other miRNAs in this figure are not examined. Also, the expression of PHB was examined, but the expression of its upstream miRNA, miR165/166, is not shown in this figure. In addition, the authors may want to briefly discuss why SNZ and TOE1 showed opposite expression patterns as AP2 in development.

Response: Thank you for this comment. We chose AP2 gene clusters, PHB and corresponding miRNAs for validation. Expression analysis of miRNA targets worked in general well, although some of the previously published RT qPCR primers actually showed poor efficiency or even are unspecific, for which new primers were designed (**Supplementary Data 8**). Despite these efforts, for some miRNAs-of-interest it was not possible to design highly efficient qPCR primer pairs for quantification of expression. Therefore, we extended the RT qPCR analyses to other miRNA genes in order to validate the miRNA-seq results. For this reason, we did not provide paired data for miRNA genes and their target TF genes in all of the cases.

One explanation for the opposite expression of AP2 vs. SMZ and TOE1 could be differences in biological functions, since SNZ and TOE1 are mainly flowering time regulators while AP2 is mainly known for its role in floral patterning. We only analyzed the expression of these three genes in flowers. In flowers, SEP3 shows distinct binding pattern in the promoters of these genes (**Supplementary Figure 8**): SMZ and TOE1 with strong SEP3 binding at stage 2, while AP2 with relatively strong binding at stages 4 and 8. This is probably due to the fact that AP2 paralogs mainly play repressive role in floral transition (Aukerman et al., *Plant Cell* 2003; Mathieu et al., *PLoS Biol.* 2009) while AP2 functions during flower development. We have added this interpretation of the expression data in the manuscript. Besides this, previous research has indicated complex negative feedback loops between these AP2-like TFs (Zhu and Helliwell, *JXB* 2011), which may also contribute to the opposite expression patterns.

Reviewer #4 (Remarks to the Author):

Feedback from Reviewer #4

For the reliability assessment of ChIP-seq data, the authors used IDR and PCA analysis indicating that the data from different conditions are comparable. The question has been well addressed.

Response: Thank you for the comment.

The authors also did the correlation analysis between gene expression changes and the binding intensity of different TFs. But the analysis of multiple target genes might generate a mixed result. The authors could include some examples of specific target genes to show the correlation which might be clearer.

Response: We thank the reviewer for this nice suggestion. We performed a similar analysis using TF-specific target genes. Only TFs with more than 200 specific target genes were included in the analysis. As shown in the figure below, similar observations are found for BLR, FLM, PI, and SVP, but not for LFY, RGA and SEP3. We have included the new analysis in the revision.

Supplementary Figure 6/7: Correlation of gene expression change and the binding intensity of different transcription factors (TFs). In the left panel, all target genes were used for the analysis. In the right panel, only TF-specific target genes were considered in the analysis. The number of genes used in the analysis is shown in parentheses.

The changes of the manuscript in regards to the previous work is acceptable.

In summary, I am happy for the paper to be published in NC after the minor points are addressed.

Response: Thank you!

Feedback to the points addressed by Reviewer #2

The authors used stem-loop RT-qPCR to validate the expression of several miRNAs and qRT-PCR to validate the expression of some known TFs. The experimental data agree well with the NGS data. It is better to include the statistical analysis addressing the significance.

Response: Thank you for this comment. We have included a statistical analysis to show the correlation between the experimental data and sequencing data (see **Supplementary Figure 9b**).

The luciferase reporter-based transient expression assays provided a strong evidence of a direct effect of SEPs binding on promoters of MIR319a and TCP4.

The PCA analysis also indicate a good overlap between different replicates from RNA-seq and CHIP-seq experiments.

Response: We thank the reviewer for the positive comments.

The questions from reviewer #2 have been satisfactorily addressed. Minor points need be addressed further.

Response: Thank you. We have addressed the points from the reviewer.